# LEARNING GRAPH REPRESENTATIONS VIA GRAPH ENTROPY MAXIMIZATION

## ABSTRACT

Graph representation learning aims to represent graphs as vectors that can be utilized in downstream tasks such as graph classification. In this work, we focus on learning diverse representations that can capture the graph information as much as possible. We propose to quantify graph information using graph entropy, where we define a probability distribution of a graph based on its node and global representations. However, computing graph entropy is NP-hard due to the complex vertex packing polytope involved in its definition. We therefore provide an approximation of graph entropy based on the Shannon entropy and the chromatic entropy. By maximizing the approximation of graph entropy through graph neural networks, we obtain informative node and graph representations. Experimental results demonstrate the effectiveness of our method in comparison to baselines in unsupervised learning and semi-supervised learning tasks.

## 1 INTRODUCTION

Graphs, such as chemical compounds (Debnath et al., 1991; Kriege & Mutzel, 2012), protein structures (Borgwardt et al., 2005), and social networks (Yanardag & Vishwanathan, 2015), represent relationships between various entities. Graph representation learning aims to convert graph-structured data into effective vector representations for various downstream tasks like graph classification. This task is nontrivial because graph data are non-Euclidean data. There have been many works of graph representation learning using the GNNs (Kipf & Welling, 2016a; Hamilton et al., 2017a; Xu et al., 2018; Veličković et al., 2017; Kipf et al., 2018; Xie & Grossman, 2018; Gilmer et al., 2017). Unsupervised graph-level representation learning is a fundamental and challenging task in this field. For example, InfoGraph (Sun et al., 2019) achieves graph-level representations by maximizing mutual information between graph-level representations and node-level representations. Graph contrastive learning (GraphCL) (You et al., 2020) and adversarial graph contrastive learning (AD-GCL) (Suresh et al., 2021) obtain graph-level representations by training GNNs to maximize the correspondence between representations of the same graph in different augmented forms. JOint Augmentation Optimization (JOAO) (You et al., 2021) is a framework that automatically and adaptively selects data augmentations for GraphCL on specific graph data using a unified bi-level min-max optimization approach. Automated Graph Contrastive Learning (AutoGCL) (Yin et al., 2022) utilizes learnable graph view generators and an auto-augmentation strategy to generate contrastive samples while preserving the most representative structures of the original graph. Graph Adversarial Contrastive Learning (GraphACL) Luo et al. (2023a) introduces a novel approach to self-supervised whole-graph representation learning by learning negative samples. InfoGCL Xu et al. (2021) delves into the transformation and transfer of graph information within the contrastive learning process, introducing an information-aware framework for graph contrastive learning. Spectral Feature Augmentation (SFA) Zhang et al. (2023) offers an efficient spectral feature augmentation method for Graph Contrastive Learning (GCL). Graph Contrastive Saliency (GCS) Wei et al. (2023) focuses on identifying semantically discriminative substructures within graphs through contrastive learning. Neighbor Contrastive Learning Augmentation (NCLA) Shen et al. (2023) enhances graph augmentation through neighbor contrastive learning. Simple Neural Networks with Structural and Semantic Contrastive Learning ($S^3$-CL) Ding et al. (2023) learns expressive node representations in a self-supervised manner. Imbalanced Node Classification (ImGCL) Zeng et al. (2023) automatically balances learned representations from GCL without labels.GRADATE Duan et al. (2023) presents a comprehensive framework for Graph Anomaly Detection, incorporating subgraph-subgraph contrast

and augmented views into multi-scale contrastive learning networks. These graph-level representation learning methods are all based on the InfoMax principle (Linsker, 1988). It is important to note that there are several other graph representation learning methods, such as VGAE (Kipf & Welling, 2016b; Hamilton et al., 2017b; Cui et al., 2020), graph embedding methods (Wu et al., 2020; Yu et al., 2021; Bai et al., 2019; Verma & Zhang, 2019), self-supervised learning (Sun et al., 2023; Liu et al., 2022b; Hou et al., 2022; Lee et al., 2022; Xie et al., 2022; Wu et al., 2021; Rong et al., 2020; Zhang et al., 2021b;a; Xiao et al., 2022), and contrastive learning (Le-Khac et al., 2020; Qiu et al., 2020; Ding et al., 2022; Xia et al., 2022; Fang et al., 2022; Trivedi et al., 2022; Han et al., 2022; Mo et al., 2022; Yin et al., 2022; Xu et al., 2021; Zhao et al., 2021; Zeng & Xie, 2021; Li et al., 2022a;b; Wei et al., 2022). Due to the page length limitation, we will not detail these methods.

In the past decades, a variety of notions of entropy have been proposed for measuring the information and complexity of graphs from different aspects (Dehmer & Mowshowitz, 2011; Dehmer, 2008). For example, the structural entropy (Mowshowitz & Dehmer, 2012) is defined on the Shannon entropy and the structural components of each node (e.g. the degree of a node). The structural entropy is widely used in GNN-based graph learning methods for measuring the topological structural information of graphs (Luo et al., 2021; Yang et al., 2023; Wang et al., 2023; Wu et al., 2022; Zou et al., 2023; Fang et al., 2021). The edge entropy is defined on the connected structure of edges and is also used to evaluate the structural information of graphs (Jiang et al., 2020; Wang et al., 2021; Grebenkina et al., 2018; Aziz et al., 2020; Luo et al., 2023b). The von Neumann entropy of a graph is defined on the graph Laplacian and is used to measure the spectral complexity of graphs (Liu et al., 2021; 2022a; Passerini & Severini, 2008; Minello et al., 2019; Ye et al., 2014; Dong et al., 2019). The Rényi entropy is a generalized information measure including various notions of entropy and is used for graph clustering tasks (Pál et al., 2010; Oggier & Datta, 2021). M-ILBO Ma et al. (2023) involves the estimation of graph dataset entropy by maximizing the Information Lower Bound (ILBO) using subsets of data samples. However, it's important to clarify that while these works use the term "graph entropy", they are not referring to the authentic Graph entropy as defined by János Körner (Körner, 1973). Graph entropy is, in fact, a fundamental concept deeply rooted in the disciplines of combinatorics and information theory, with a rich history. In 1948, Claude E. Shannon laid the foundations of information theory and introduced the concept of channel capacity (Shannon, 1948). Subsequently, in 1973, János Körner introduced the concept of graph entropy, which serves as a measure of the information that can be effectively communicated over a noisy channel (Körner, 1973). While graph entropy finds its origins in information theory, it also finds utility in quantifying the information within a set, especially when some pairs of elements share common information (Bouchon et al., 1988). In 1979, László Lovász introduced orthonormal representations with the aim of analyzing a graph's Shannon capacity (Lovász, 1979). These representations comprise sets of vertex representation vectors, allowing for the possibility that adjacent vertices may share common information. This property aligns with the combinatorics definition of graph entropy. Nevertheless, the computation of graph entropy becomes a computationally challenging task due to the complex vertex packing polytope involved in its definition.

While graph entropy has found success in the realms of combinatorics and information theory, it remains relatively unexplored within the field of graph learning. In this study, we introduce a novel approach called Graph Entropy Maximization (GeMax) for graph representation learning, marking the first instance of its application in this context. Our approach begins with two key insights: firstly, the necessity of employing orthonormal representations for nodes, which enables direct measurement of the information contained in graph representations using graph entropy. Secondly, we establish a probability distribution for a graph by incorporating its nodes' representations and a global graph representation learned through two separate graph neural networks. Recognizing the computational challenges associated with graph entropy, we propose an approximation method that leverages Shannon entropy and chromatic entropy. This approximation leads to a max-min optimization problem, which we address through alternating minimization techniques. Through extensive experimentation, we validate the competitiveness of our proposed methods against various baselines in both unsupervised graph-level and node-level learning tasks. In summary, our contributions in this work can be categorized into three main areas.

- We introduce a novel method, Graph Entropy Maximization (GeMax), for graph representation learning, marking the inaugural exploration of Körner's graph entropy within the graph learning community.

- Our framework for employing Körner's graph entropy consists of two key components: firstly, the adoption of orthonormal representations for nodes, facilitating direct quantification of information within graph representations through graph entropy. Secondly, the establishment of a probability distribution for a given graph.
- Additionally, we present an approximation technique to compute graph entropy, leveraging both Shannon entropy and chromatic entropy.

## 2 PRELIMINARIES

In this section, we introduce the definitions of orthonormal representations, graph entropy and chromatic entropy. The main notations used in this paper are shown in Table 5 of Appendix A.1.

### 2.1 ORTHONORMAL REPRESENTATIONS

Given a graph $G = (V, E)$, if two vertices share some information and may be confused in communication, they are adjacent with an edge. In contrast, if there is no common information between two vertices, they should be non-adjacent. Based on this intuition, László Lovász introduced the orthonormal representations of a graph (Lovász, 1979). If two vertices are non-adjacent, Lovász argued that their representation vectors should be orthogonal to each other, indicating that there is no common information between them.

**Definition 2.1** (Set of orthonormal representations (Lovász, 1979)). Given a graph $G = (V, E)$, we use a unit vector $\boldsymbol{z}_i \in \mathbb{R}^d$ to denote the $d$-dimensional representation of vertex $i$. Then the set of orthonormal representations of $G$ is defined as

$$\mathbb{T}(G) := \{\boldsymbol{Z} \in \mathbb{R}^{d \times n} : \|\boldsymbol{z}_i\|_2 = 1, i = 1, 2, \ldots, n; \ \boldsymbol{z}_i^\top \boldsymbol{z}_j = 0, \ \forall (i,j) \notin E\}. \tag{1}$$

### 2.2 GRAPH ENTROPY

Graph entropy is a fundamental property of a probabilistic graph, first introduced by Körner (1973). Though there exist several equivalent definitions of graph entropy, we focus on its combinatorial definition which is based on the vertex packing polytope $\text{VP}(G)$. In graph theory, the independent set is a set of vertices of $\mathcal{G}$ where no two vertices are adjacent. Let $\boldsymbol{B} = [\boldsymbol{b}_1, ..., \boldsymbol{b}_{N_b}] \in \{0, 1\}^{|V| \times N_b}$ be the indicator matrix of independent sets of $G$, where $N_b$ is number of independent sets of $G$ and the $i$-th column $\boldsymbol{b}_i$ is the indicator vector of the $i$-th independent set. For example, if $G$ is a pentagon in Figure 1, we have $N_b = 10$ and $\boldsymbol{b}_6 = [1, 0, 1, 0, 0]^\top$ indicates the vertex subset $\{v_1, v_3\}$ is the 6-th independent set of $G$. The vertex packing polytope $\text{VP}(G)$ is defined as follows.

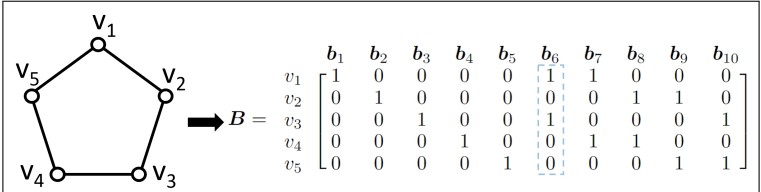

Figure 1: The indicator matrix $\boldsymbol{B}$ of independent sets of a pentagon

**Definition 2.2** (Vertex packing polytope). Given a graph $G$ with vertex set $V$, the vertex packing polytope $\text{VP}(G)$ of $G$ is the convex hull of the indicator vectors of its independent sets. More precisely, let $\boldsymbol{B} \in \{0, 1\}^{|V| \times N_b}$ be the indicator matrix of independent sets of $G$ and $\boldsymbol{\lambda} \in \mathbb{R}_+^{N_b}$ be a vector, then $\text{VP}(G)$ is defined as follows

$$\text{VP}(G) := \left\{ \boldsymbol{a} \in \mathbb{R}^{|V|} : \boldsymbol{a} = \boldsymbol{B}\boldsymbol{\lambda}, \text{ with } \boldsymbol{\lambda} \geq 0, \sum_{i=1}^{N_b} \lambda_i = 1 \right\}. \tag{2}$$

Let $(G, P)$ be a probabilistic graph with respect to probability distribution $P$ on its vertex set $V$, i.e., $P = \{P_1, P_2, ..., P_n\}$ and $P_i$ is the probability density of the vertex $i$. The graph entropy is usually denoted as $H_k(G, P)$, named after János Körner. Based on $\text{VP}(G)$, we have

**Definition 2.3** (Graph entropy Körner (1973)). Given a probabilistic graph $(G, P)$ with $|V| = n$, the entropy of $G$ with respect to $P$ is defined as

$$H_k(G, P) := \min_{\boldsymbol{a} \in \text{VP}(G)} \sum_{i=1}^{n} -P_i \log(a_i), \tag{3}$$

where $\boldsymbol{a}$ is a vector in the vertex packing polytope $\text{VP}(G)$ and $a_i$ is the $i$-th element of vector $\boldsymbol{a}$.

In information theory, the graph entropy is a measure of the maximal information rate of communicating over a noisy channel. This notion has been extended to combinatorics as follows.

**Corollary 2.4** ((Bouchon et al., 1988)). *In combinatorics, graph entropy can be used to measure the amount of information contained in a set where some pairs of elements contain common information.*

It is known that a graph is a set and its vertices are the elements, where adjacent vertices contain common information. Thus, according to Corollary 2.4, **graph entropy can be used to measure the amount of information contained in the orthonormal representations of a graph**.

### 2.3 CHROMATIC ENTROPY

We introduce the definition of the chromatic entropy (Alon & Orlitsky, 1996). A coloring of graph $G$ is the process of assigning colors to vertices such that no adjacent vertices share the same color. Let $\boldsymbol{\pi} = \{\mathbb{C}_1, ..., \mathbb{C}_K\}$ be a coloring with $K$ colors on $G$, i.e., $\boldsymbol{\pi}$ is a partition of the vertex set $V$ and $\mathbb{C}_k$ is the set of all the vertices with the $k$-th color class. The entropy of a coloring $\boldsymbol{\pi}$ on a probabilistic graph $(G, P)$ is denoted by $H_c(G, P, \boldsymbol{\pi})$ and is defined as follows.

**Definition 2.5** (Entropy of a coloring). Given a probabilistic graph $(G, P)$ with $|V| = n$ and a coloring $\boldsymbol{\pi} = \{\mathbb{C}_1, ..., \mathbb{C}_K\}$, the probability distribution on the coloring $\boldsymbol{\pi}$ is defined by summing up the probability density of the vertices with the same color, i.e,

$$P(\mathbb{C}_k) := \sum_{v \in \mathbb{C}_k} P_v, \quad \forall k \in \{1, .., K\}, \tag{4}$$

where $P_v$ is the probability density of vertex $v$. The entropy of a coloring $\boldsymbol{\pi}$ is defined as

$$H_c(G, P, \boldsymbol{\pi}) := \sum_{k=1}^{K} -P(\mathbb{C}_k) \log P(\mathbb{C}_k). \tag{5}$$

Let $\boldsymbol{\Pi}(G)$ be the set of all possible coloring $\boldsymbol{\pi}$ of graph $G$. Then the chromatic entropy is defined as

**Definition 2.6** (Chromatic entropy Alon & Orlitsky (1996)). The chromatic entropy of a probabilistic graph $(G, P)$ is the lowest entropy among all possible colorings of the graph, i.e.,

$$H_\chi(G, P) := \min\{H_c(G, P, \boldsymbol{\pi}) : \boldsymbol{\pi} \in \boldsymbol{\Pi}(G)\}. \tag{6}$$

Let $\chi_H(G, P)$ be the minimum number of colors for $H_\chi(G, P)$ and $\Delta(G)$ be the maximum degree of a vertex in $G$. It follows that

**Corollary 2.7** ((Rezaei, 2013)). $\chi_H(G, P) \leq \Delta(G) + 1$.

### 2.4 LOWER AND UPPER BOUNDS OF GRAPH ENTROPY

Let $\alpha(G)$ be the independence number which is the size of the maximum independent set of graph $G$. The lower bound and upper bound of graph entropy $H_k(G, P)$ are introduced as follows (Boreland, 2018; Alon & Orlitsky, 1996; Cardinal et al., 2004; 2005).

**Theorem 2.8** (Lower and upper bounds of graph entropy). *Given a probabilistic graph $(G, P)$, the lower and upper bounds of the graph entropy $H_k(G, P)$ are as follows,*

$$H(P) - \log \alpha(G) \leq H_k(G, P) \leq H_\chi(G, P), \tag{7}$$

*where $H(P)$ is the Shannon entropy and $H_\chi(G, P)$ is the chromatic entropy in Definition 2.6.*

**Corollary 2.9** ((Boreland, 2018)). *The equality of the lower bound in Theorem 2.8 holds if and only if there exists a vector $\boldsymbol{h} \in VP(G)$ satisfying $h_i = P_i \alpha(G)$ for $i = 1, 2, ..., n$.*

**Corollary 2.10** ((Rezaei, 2013)). *The equality of the upper bound in Theorem 2.8 holds when $G$ is an empty graph or a complete graph.*

## 3  MOTIVATION AND PROBLEM SETUP

Our motivation revolves around the utilization of graph entropy as a means to quantify the information contained within graph representations. Specifically, we aim to identify representations that can capture the most significant information, essentially those with the maximum graph entropy. To ensure that node representations accurately capture structural information, we recommend the adoption of orthonormal representations as defined in Definition 2.1. This choice is rooted in the fact that graph entropy can directly serve as a measure of information within orthonormal representations, as indicated by Corollary 2.4. However, an issue arises with the involvement of the probability of each node in Definition 2.3, as it lacks a clear definition. To address this, we introduce the concept of a graph-level representation $\boldsymbol{g}$ and associate a Gaussian distribution $P$ with the vertex set $V$, using $\boldsymbol{g}$ as the mean. Let $\boldsymbol{Z} = [\mathbf{z}_1, \mathbf{z}_2, \ldots, \mathbf{z}_n]$, where $\mathbf{z}_i$ represents the node representation of node $i$ in graph $G$. We define $P = P(\mathbf{g}, \boldsymbol{Z}) = P1(\mathbf{g}, \boldsymbol{Z}), ..., P_n(\mathbf{g}, \boldsymbol{Z})$, and establish the following:

$$P_i(\mathbf{g}, \boldsymbol{Z}) := \frac{\exp(-\|\boldsymbol{z}_i - \mathbf{g}\|_2^2)}{\sum_{j \in V} \exp(-\|\boldsymbol{z}_j - \mathbf{g}\|_2^2)}, \ \ \forall \, i = 1, 2, ..., n. \tag{8}$$

Then we formulate the learning of orthonormal representation with maximum graph entropy as

$$\max_{\mathbf{g}, \boldsymbol{Z} \in \mathbb{T}(G)} \ \min_{\boldsymbol{a} \in \mathrm{VP}(G)} \sum_{i=1}^{n} -P_i(\mathbf{g}, \boldsymbol{Z}) \log(a_i) \tag{9}$$

Given a set of $N$ graphs $\mathcal{G} = \{G_1, G_2, \ldots, G_N\}$ drawn from some unknown distribution $\mathcal{D}$ in $\mathbb{G}$, we want to learn a model $F : \mathbb{G} \to \mathbb{R}^d \times \mathbb{R}^{d \times n}$ to represent each graph as a vector and represent its vertices as vectors, i.e., $(\mathbf{g}_j, \boldsymbol{Z}_j) = F(G_j)$, where $F$ should capture the important information of the underlying distribution $\mathcal{D}$ and $\mathbf{g}_1, \mathbf{g}_2, \ldots, \mathbf{g}_N$ should be useful in downstream tasks such as graph classification. Based on (9), we propose to solve the following problem

$$\max_{F \in \mathcal{F}} \ \mathbb{E}_{G \sim \mathcal{D}} \left[ \min_{\boldsymbol{a} \in \mathrm{VP}(G)} \sum_{i=1}^{n} -P_i(\mathbf{g}, \boldsymbol{Z}) \log(a_i) \right] \tag{10}$$
$$\text{s.t. } (\mathbf{g}, \boldsymbol{Z}) = F(G), \ \boldsymbol{Z} \in \mathbb{T}(G).$$

The problem equation 10 is our Graph Entropy Maximization (GeMax) problem for graph representation learning.

## 4  METHODOLOGY FOR MAXIMIZING GRAPH ENTROPY.

### 4.1  APPROXIMATION OF GRAPH ENTROPY

Directly solving the GeMax problem equation 10 is NP-hard due to the complex vertex packing polytope $\mathrm{VP}(G)$ in Definition 2.3. In information theory and combinatorics, graph entropy is typically applied for theoretical analysis rather than practical computation. In this study, we approximate the value of graph entropy using its lower and upper bounds in Theorem 2.8. We can maximize the lower bound of graph entropy $H_k(G, P)$ to estimate the solution of the problem in equation 9, i.e.,

$$\max_{\mathbf{g}, \boldsymbol{Z}} \ H(P(\mathbf{g}, \boldsymbol{Z})) - \log \alpha(G), \ \text{ s.t. } \boldsymbol{Z} \in \mathbb{T}(G). \tag{11}$$

Since the independence number $\alpha(G)$ is a constant of a given $G$, the optimization in equation 11 is actually learning the orthonormal representations via maximizing the Shannon entropy $H(P(\mathbf{g}, \boldsymbol{Z}))$. Suppose the representations $(\mathbf{g}^*, \boldsymbol{Z}^*)$ yield the maximum of equation 11 where $\boldsymbol{Z}^* \in \mathbb{T}(G)$. Based on Corollary 2.9, if there exists a vector $\boldsymbol{h} \in \mathrm{VP}(G)$ satisfying $h_i = P_i \alpha(G)$ on the vertex set $V$, the equality of the lower bound in equation 7 holds and we have $H(P(\mathbf{g}^*, \boldsymbol{Z}^*)) - \log \alpha(G) = H_k(G, P(\mathbf{g}^*, \boldsymbol{Z}^*))$. That is, the graph entropy maximization in equation 9 is equivalent to the Shannon entropy maximization equation 11 in this case. However, the equality of the lower bound in equation 7 is not guaranteed to hold for an arbitrary probabilistic graph $(G, P)$. We need to approximate the graph entropy $H_k(G, P)$ for more general cases. According to Theorem 2.8, $H_k(G, P)$ can be represented as a convex combination of the lower and upper bounds as follows.

**Corollary 4.1.** *There exits $0 \le \mu \le 1$ such that*

$$H_k(G, P) = \mu(H(P) - \log \alpha(G)) + (1 - \mu) H_\chi(G, P), \tag{12}$$

Given that $\alpha(G)$ is a constant, the orthonormal representations learning problem (9) can be rewritten according to Corollary 4.1 as,

$$\max_{\mathbf{g}, \mathbf{Z}} \ \mu H(P(\mathbf{g}, \mathbf{Z})) + (1 - \mu) H_\chi(G, P(\mathbf{g}, \mathbf{Z})), \ \text{s.t.} \ \mathbf{Z} \in \mathbb{T}(G), \tag{13}$$

That is, maximizing the graph entropy $H_k(G, P)$ is equivalent to maximizing the Shannon entropy $H(P)$ and chromatic entropy $H_\chi(G, P)$ simultaneously. The motivation for maximizing Shannon entropy $H(P)$ is to maximize the lower bound of graph entropy $H_k(G, P)$ in (11). Suppose the coloring $\boldsymbol{\pi}^* = \{\mathbb{C}_1^*, ..., \mathbb{C}_K^*\}$ is the coloring aligns with chromatic entropy $H_\chi(G, P)$ where $\mathbb{C}_k^*$ is the set of all nodes in vertex set $V$ with the $k$-th color. The color set $\mathbb{C}_k^*$ is actually an independent set where the orthonormal representations are orthogonal to each other. Maximizing chromatic entropy $H_\chi(G, P)$ is actually to maximize the information contained in independent sets with respect to coloring $\boldsymbol{\pi}^*$. Based on the chromatic entropy Definition 2.6, we reformulate problem (10) as the following constrained max-min problem:

$$\max_{F \in \mathcal{F}} \ \mathbb{E}_{G \sim \mathcal{D}} \left[ \min_{\boldsymbol{\pi}} \mu H(P(\mathbf{g}, \mathbf{Z})) + (1 - \mu) H_c(G, P(\mathbf{g}, \mathbf{Z}), \boldsymbol{\pi}) \right]$$
$$\text{s.t.} \ (\mathbf{g}, \mathbf{Z}) = F(G), \ \mathbf{Z} \in \mathbb{T}(G), \ \boldsymbol{\pi} \in \boldsymbol{\Pi}(G). \tag{14}$$

## 4.2 Representation Learning via GNNs

**Parameterization** Developing an optimization algorithm to solve the problem 14 is difficult. Thus we learn a GNN model from a set of graphs $\mathcal{G}$ to solve the problem 14 and find the representations of graphs in $\mathcal{G}$. Denote $\mathbb{A}$ as the space of adjacent matrix $\boldsymbol{A}$ and $\mathbb{X}$ as the space of node feature matrix $\boldsymbol{X}$. Let $F_g(\cdot, \cdot; \boldsymbol{\theta}) : \mathbb{A} \times \mathbb{X} \to \mathbb{R}^d$ be a GNN with parameter $\boldsymbol{\theta}$ for graph-level representation learning and $F_Z(\cdot, \cdot; \boldsymbol{\phi}) : \mathbb{A} \times \mathbb{X} \to \mathbb{R}^d$ be another GNN with parameters $\boldsymbol{\phi}$ for node-level representation learning. For $G \in \mathcal{G}$ with adjacency matrix $\boldsymbol{A}$ and feature matrix $\boldsymbol{X}$, we obtain

$$\mathbf{g}^\theta = F_g(\boldsymbol{A}, \boldsymbol{X}; \boldsymbol{\theta}) \ \text{and} \ \boldsymbol{Z}^\phi = F_Z(\boldsymbol{A}, \boldsymbol{X}; \boldsymbol{\phi}). \tag{15}$$

Let $\boldsymbol{C} = [\boldsymbol{c}_1, ..., \boldsymbol{c}_n]^\top \in \mathbb{R}^{n \times K}$ be a color probability matrix where $\boldsymbol{c}_i = [c_{i(1)}, ..., c_{i(K)}]^\top$ and $c_{i(k)}$ is the probability of coloring node $i$ with color $k$, i.e., $i \in \mathbb{C}_k$. According to Corollary 2.7, we can directly set $K = \Delta(G) + 1$. Letting $F_c(\cdot, \cdot; \boldsymbol{\psi}) : \mathbb{A} \times \mathbb{X} \to \mathbb{R}^{n \times K}$ be a GNN with parameter $\boldsymbol{\psi}$ for learning coloring, we have

$$\boldsymbol{C}^\psi = F_c(\boldsymbol{A}, \boldsymbol{X}; \boldsymbol{\psi}), \tag{16}$$

where $F_c$ should ensures that $0 \le c_{i(k)}^\psi \le 1$ and $\sum_{k=1}^K c_{i(k)}^\psi = 1$. Thus the output activation function of $F_c$ should be a *softmax* function. The coloring $\boldsymbol{\pi}$ can be parameterized by $\boldsymbol{\psi}$ as follows

$$\boldsymbol{\pi}^\psi = \{\mathbb{C}_1^\psi, ..., \mathbb{C}_K^\psi\} \ \text{where} \ i \in \mathbb{C}_k^\psi \ \text{if} \ k = \operatorname*{argmax}_{q=1,...,K} c_{i(q)}^\psi \ \forall i \in V. \tag{17}$$

**Regularizations and Losses** Instead of solving constrained optimization, we propose to solve unconstrained optimization with regularization, which is much easier. Based on Definition 2.1, we propose the orthonormal loss $\ell_{\text{orth}}(G; \boldsymbol{\phi})$ for regularization of $\boldsymbol{Z} \in \mathbb{T}(G)$ as follows

$$\ell_{\text{orth}}(G; \boldsymbol{\phi}) = \frac{1}{2} \left\| \left( \boldsymbol{Z}^\phi (\boldsymbol{Z}^\phi)^\top - \boldsymbol{I}_n \right) \odot \boldsymbol{M} \right\|_F^2, \tag{18}$$

where $\boldsymbol{M} = \mathbf{1}_{n \times n} - \boldsymbol{A}$ is a binary mask matrix and $\odot$ is the Hadamard product. We also introduce the coloring loss $\ell_c(G; \boldsymbol{\psi})$ for regularization of $\boldsymbol{\pi} \in \boldsymbol{\Pi}(G)$ as follows

$$\ell_c(G; \boldsymbol{\psi}) = \frac{1}{2} \left\| \left( \boldsymbol{C}^\psi (\boldsymbol{C}^\psi)^\top \right) \odot \boldsymbol{A} \right\|_F^2. \tag{19}$$

Therefore, the overall objective function of the regularized rather than the constrained max-min problem 14 on the dataset $\mathcal{G}$ is formulated as

$$J(\mathcal{G}; \boldsymbol{\theta}, \boldsymbol{\phi}, \boldsymbol{\psi}) := \sum_{j=1}^N \left\{ \mu H(P(\mathbf{g}_j^\theta, \boldsymbol{Z}_j^\phi)) + (1 - \mu) H_c(G_j, P(\mathbf{g}_j^\theta, \boldsymbol{Z}_j^\phi), \boldsymbol{\pi}^\psi) \right.$$
$$\left. - \gamma \ell_{\text{orth}}(G_j; \boldsymbol{\phi}) + \eta \ell_c(G_j; \boldsymbol{\psi}) \right\}, \tag{20}$$

where $\gamma$ and $\eta$ are positive hyperparameters for regularization.

**Iterative Optimization** Given a coloring $\boldsymbol{\pi}^*$, the representations updating sub-problem is to optimize $\boldsymbol{\theta}$ and $\boldsymbol{\phi}$ with a fixed $\boldsymbol{\psi}^*$ as follows

$$\theta^*, \phi^* = \underset{\theta, \phi}{\arg\max}\, J(\mathcal{G}; \boldsymbol{\theta}, \boldsymbol{\phi}, \boldsymbol{\psi}^*), \tag{21}$$

Given graph representations $(\mathbf{g}^*, \boldsymbol{Z}^*)$, the coloring updating sub-problem is to optimize $\boldsymbol{\psi}$ with fixed $\boldsymbol{\theta}^*$ and $\boldsymbol{\phi}^*$ as follows

$$\boldsymbol{\psi}^* = \underset{\boldsymbol{\psi}}{\arg\min}\, J(\mathcal{G}; \boldsymbol{\theta}^*, \boldsymbol{\phi}^*, \boldsymbol{\psi}), \tag{22}$$

It is worth noting that we have relaxed the original constrained optimization to a regularized unconstrained optimization. The solution cannot satisfy the constraints exactly. To ensure the constraints, one may consider using the exact penalty method to solve the problem, which is just increasing $\gamma$ and $\eta$ gradually in the iterative optimization. But we have found that the representations given the regularized optimization are as good as those given by the constrained optimization. The comparison experiments between regularized and constrained optimization are in Appendix A.11.

**Error Bound** Let $\hat{\mu}$ be an estimator of $\mu$ and $\hat{H}_k(G, P)$ be the approximation of $H_k(G, P)$ with respect to $\hat{\mu}$. Let $\delta$ be the upper bound of error bound of our approximation. Let $\chi(G)$ be chromatic number of graph $G$ which is the smallest number of colors needed to color the vertices.

**Corollary 4.2.** *Let* $\epsilon = \frac{\log \chi(G)}{1+\delta} + \log \alpha(G)$, *if* $H(P) \geq \epsilon$, *we have* $\frac{|H_k(G,P) - \hat{H}_k(G,P)|}{H_k(G,P)} \leq \delta$.

This means that if the Shannon entropy is larger than $\epsilon$, the error bound of approximation is less than $\delta$. The proof of Corollary 4.2 is in Appendix A.2. In the $t$ iteration, the average Shannon entropy of the dataset $\mathcal{G}$ is defined as

$$\bar{H}(\mathcal{G}; t) := \frac{1}{N} \sum_{j=1}^{N} H(P(\mathbf{g}_j^{\theta^t}, \boldsymbol{Z}_j^{\phi^t})). \tag{23}$$

Based on Corollary 4.2, if $\bar{H}(\mathcal{G}; t) < \epsilon$ and $\mu^t + 0.01 \leq 1$, we use $\mu^{t+1} = \mu^t + 0.01$ to increase the average Shannon entropy of $\mathcal{G}$ for a more exact approximation.

---

**Algorithm 1** Iterative algorithm for solving the regularized max-min problem (14)

1: **Initialization:** $\boldsymbol{\theta}^0, \boldsymbol{\phi}^0, \boldsymbol{\psi}^0, \mu^0 = 0.5, \gamma = 0.5, \eta = 0.5, \epsilon$ (e.g., $0.3 \log n$), $\varepsilon$ (e.g., $0.01$).
2: **repeat**
3: $\quad \theta^{t+1}, \phi^{t+1} = \arg\max_{\theta, \phi} J(\mathcal{G}; \boldsymbol{\theta}, \boldsymbol{\phi}, \boldsymbol{\psi}^t)$
4: $\quad \boldsymbol{\psi}^{t+1} = \arg\min_{\boldsymbol{\psi}} J(\mathcal{G}; \boldsymbol{\theta}^{t+1}, \boldsymbol{\phi}^{t+1}, \boldsymbol{\psi})$
5: $\quad$ **if** $\bar{H}(\mathcal{G}; t) < \epsilon$ and $\mu^t + 0.01 \leq 1$ **then** $\mu^{t+1} = \mu^t + 0.01$ **else** $\mu^{t+1} = \mu^t$
6: **until** $|J(\mathcal{G}; \boldsymbol{\theta}^{t+1}, \boldsymbol{\phi}^{t+1}, \boldsymbol{\psi}^{t+1}) - J(\mathcal{G}; \boldsymbol{\theta}^t, \boldsymbol{\phi}^t, \boldsymbol{\psi}^t)| \leq \varepsilon$

---

### 4.3 Architecture and Generalizations

The architecture of our GeMax method is in Figure 2 of Appendix A.3. The graph representations learning functions $F_g(\cdot, \cdot; \boldsymbol{\theta})$ and $F_Z(\cdot, \cdot; \boldsymbol{\phi})$ are not confined to any specific GNN models or graph data; rather, it offers a versatile approach across various contexts. For example, we can use one of the InfoMax-based models (e.g. InfoGraph (Sun et al., 2019) or GraphCL (You et al., 2020)) to model graph representation learning $F_g$ and $F_Z$ for our GeMax (see Figure 3 in Appendix A.3). In conclusion, our graph entropy maximization principle is parallel to other unsupervised graph learning principles such as InfoMax principle (Linsker, 1988).

## 5 Experiment

In this section, we will evaluate the effectiveness of our Graph entropy Maximization (GeMax) method on graph-level and node-level tasks. The statistics of graph datasets used in experiments are in Table 6 and Table 7 of Appendix A.4. In Appendix A.5, we introduce our main baseline InfoMax (Linsker, 1988). We provide the experimental settings of node-level learning in Appendix

A.7. In Appendix A.8, we conduct sensitivity analysis on all the hyperparameters and provide some recommendations for parameter settings. In Appendix A.9, we conduct an ablation study to analyze the importance of each part in our graph entropy maximization methods. In Appendix A.10, we analyze the convergence of our iterative algorithm 1. In Appendix A.11, we compare the regularized and constrained optimization of problem equation 14. In Appendix A.12, we report the time cost of our methods on different tasks and datasets.

## 5.1 GRAPH-LEVEL REPRESENTATIONS LEARNING

For unsupervised graph-level learning, those InfoMax based methods are the most current and influential methods spanning from 2019 to 2022, each boasting high citations on Google Scholar (see Table 8). Besides InfoMax, there are few works on other graph-level representation learning principles such as graph information bottleneck (GIB) Wu et al. (2020) and the subgraph information bottleneck (SIB) Yu et al. (2021). But they are not suitable for unsupervised graph learning (see Appendix A.5). To ensure fair comparisons, we follow the neural network architectures of InfoMax methods and replace the InfoMax objective with our GeMax objective in equation 20. Our baselines include three kernel methods (e.g. graphlet kernel (GL) (Shervashidze et al., 2009), Weisfeiler-Lehman sub-tree kernel (WL) (Shervashidze et al., 2011), deep graph kernel (DGK) (Yanardag & Vishwanathan, 2015)), two traditional graph embedding methods (e.g. node2vec (Grover & Leskovec, 2016), and graph2vec (Narayanan et al., 2017)), and five InfoMax based methods (e.g InfoGraph (Sun et al., 2019), GraphCL (You et al., 2020), AD-GCL (Suresh et al., 2021), JOAO (You et al., 2021), AutoGCL (Yin et al., 2022)).

**Unsupervised learning**  Following (Sun et al., 2019; You et al., 2021; Yin et al., 2022), we train a graph representation model on unlabeled data to obtain graph representations and use these representations and graph labels to train a SVM classifier. Our experimental setup follows AutoGCL (Yin et al., 2022). Specifically, they use a 5-layer GIN Xu et al. (2018) with hidden size 128 as the representation model, shown in Figure 4. The model is trained with a batch size of 128 and a learning rate of 0.001. For those contrastive learning methods (e.g., JOJOv2 and AutoGCL), they use 30 epochs of contrastive pre-training under the naive strategy. We repeat the experiments for 10 times with different random seeds. In each time, we perform 10-fold cross-validation on each dataset. The hyperparameters of Algorithm 1 are $\mu^0 = 0.5, \gamma = 0.5, \eta = 0.5, \epsilon = 0.3 \log n, \tau = 0.01$. We also conduct sensitivity analysis in Appendix A.8 to study how different hyperparameters affect the results. The results are reported in Table 14.

**Semi-supervised Learning**  Following (Hu et al., 2019; You et al., 2021; Yin et al., 2022), we compare our GeMax methods with InfoMax-based methods in semi-supervised learning tasks. The semi-supervised learning objective of InfoMax method is shown in equation 29 of Appendix A.5. To ensure fair comparisons, we replace the InfoMax objective with our GeMax objective in equation 20 while keeping other settings unchanged. Following the settings of AutoGCL (Yin et al., 2022), we employ a 10-fold cross-validation on each dataset. For each fold, we use 80% of the total data as the unlabeled data, 10% as labeled training data, and 10% as labeled testing data. The classifier for labeled data is a ResGCN (Chen et al., 2019) with 5 layers and a hidden size of 128. The hyperparameters of Algorithm 1 are $\mu^0 = 0.5, \gamma = 0.5, \eta = 0.5, \epsilon = 0.3 \log n, \tau = 0.01$. We repeat each experiment 10 times and report the average accuracy in Table 2.

**Significance analysis**  Our GeMax method achieves the best performance on all datasets. By replacing the InfoMax objective with GeMax objective, the performance of the five graph representation learning methods can be improved, which demonstrates the effectiveness of our GeMax method. we apply the paired t-test on the mean scores over the datasets to show the significance of our improvement over baselines. A p-value less than 0.05 indicates a significant difference. The results in Table 3 demonstrates the significance of gains given by our methods.

## 5.2 UNSUPREVISED NODE-LEVEL LEARNING

As mentioned above, the orthonormal representations can be used for graph reconstruction. We compare GeMax methods with VGAE Kipf & Welling (2016b), ARGA Pan et al. (2018), GIC Mavromatis & Karypis (2020) and LGAE Salha et al. (2021) in edge prediction tasks. Following

Table 1: Performance (ACC) of unsupervised learning. The baseline results are from AutoGCL (Yin et al., 2022) and JOAO (You et al., 2021). The **bold**, blue and green numbers denote the best, second best and third best performances respectively, which also applies to Table 2 and Table 4

| methods and principles | | MUTAG | PROTEINS | DD | NCI1 | COLLAB | IMDB-B | REDDIT-B | REDDIT-M5K |
|---|---|---|---|---|---|---|---|---|---|
| kernels | GL | 81.66±2.11 | - | - | - | - | 65.87±0.98 | 77.34±0.18 | 41.01± 0.17 |
| | WL | 80.72±3.00 | 72.92±0.56 | - | 80.01±0.50 | - | 72.30±3.44 | 68.82±0.41 | 46.06± 0.21 |
| | DGK | 87.44±2.72 | 73.30±0.82 | - | 80.31±0.46 | - | 66.96±0.56 | 78.04±0.39 | 41.27±0.18 |
| vector embedding | node2vec | 72.63±10.20 | 57.49±3.57 | - | 54.89±1.61 | - | - | - | - |
| | graph2vec | 83.15±9.25 | 73.30±2.05 | - | 73.22±1.81 | - | 71.10±0.54 | 75.78±1.03 | 47.86±0.26 |
| InfoGraph | InfoMax | 89.01±1.13 | 74.44±0.31 | 72.85±1.78 | 76.20±1.06 | 70.65±1.13 | 73.03±0.87 | 82.50±1.42 | 53.46±1.03 |
| | GeMax | **91.13±1.70** | 75.77±1.26 | 74.16±1.65 | 79.24±1.43 | 72.57±1.74 | **74.59±1.53** | 85.53±1.92 | 55.21±1.69 |
| GraphCL | InfoMax | 86.80±1.34 | 74.39±0.45 | 78.62±0.40 | 77.87±0.41 | 71.36±1.15 | 71.14±0.44 | 89.53±0.84 | 55.99±0.28 |
| | GeMax | 90.36±1.69 | **76.86±1.62** | **79.25±1.53** | 78.72±1.79 | 73.43±1.62 | 73.12±1.25 | **91.47±1.74** | 56.25±1.53 |
| AD-GCL | InfoMax | 87.13±1.56 | 73.59±0.65 | 74.49±0.52 | 69.67±0.51 | 73.32±0.61 | 71.57±1.01 | 85.52±0.79 | 53.00±0.82 |
| | GeMax | 89.68±1.47 | 74.52±1.71 | 77.58±1.41 | 76.35±1.62 | **74.83±1.79** | 73.52±1.45 | 88.03±1.62 | 55.03±1.54 |
| JOAOv2 | InfoMax | 86.91±1.01 | 71.25±0.85 | 66.91±1.75 | 72.99±0.75 | 70.40±2.21 | 71.60±0.86 | 78.35±1.38 | 55.57±2.86 |
| | GeMax | 88.33±1.58 | 74.63±1.87 | 72.60±1.35 | 75.36±1.42 | 71.68±1.67 | 72.21±1.72 | 81.68±1.40 | **57.17±1.67** |
| AutoGCL | InfoMax | 88.64±1.08 | 75.80±0.36 | 77.57±0.60 | 82.00±0.29 | 70.12±0.68 | 73.30±0.40 | 88.58±1.49 | 56.75±0.18 |
| | GeMax | 90.85±1.28 | 76.23±1.29 | 78.36±1.51 | **83.21±1.34** | 72.39±1.57 | 74.05±1.79 | 90.42±1.31 | 56.81±1.85 |

Table 2: Performance (ACC) of semi-supervised learning.

| | methods | NCI1 | PROTEINS | DD | COLLAB | REDDIT-B | REDDIT-M5K | GITHUB |
|---|---|---|---|---|---|---|---|---|
| GraphCL | InfoMax | 74.63±0.25 | 74.17±0.34 | 76.17±1.37 | 74.23±0.21 | 89.11±0.19 | 52.55±0.45 | 65.81±0.79 |
| | GeMax | 75.49±1.76 | 75.39±1.58 | 77.61±1.29 | 76.57±1.72 | **91.45±1.57** | 54.61±1.70 | 66.78±1.53 |
| AD-GCL | InfoMax | 75.18±0.31 | 73.96±0.47 | 77.91±0.73 | 75.82±0.26 | 90.10±0.15 | 53.49±0.28 | 64.17±1.38 |
| | GeMax | 76.27±1.44 | 75.21±1.78 | **78.52±1.53** | 76.92±1.81 | 91.32±1.67 | **54.88±1.21** | 65.52±1.45 |
| JOAOv2 | InfoMax | 74.86±0.39 | 73.31±0.48 | 75.81±0.72 | 75.53±0.18 | 88.79±0.65 | 52.71±0.28 | 66.66±0.60 |
| | GeMax | **76.05±1.23** | 74.52±1.61 | 76.30±1.54 | 76.25±1.24 | 90.05±1.76 | 54.07±1.52 | 66.47±1.93 |
| AutoGCL | InfoMax | 73.75±2.25 | 75.65±2.40 | 77.50±4.41 | 77.16±1.48 | 79.80±3.47 | 49.91±2.70 | 62.46±1.51 |
| | GeMax | 75.12±1.19 | **76.75±1.83** | 78.36±1.37 | **78.93±1.80** | 87.26±1.68 | 52.76±1.74 | **67.31±1.64** |

Table 3: Significance analysis of improvement. We report the p-values of paired t-test.

| task (principles) | InfoGraph | GraphCL | AD-GCL | JOAOv2 | AutoGCL |
|---|---|---|---|---|---|
| unsupervised (InfoMax vs GeMax) | 0.0005 | 0.0029 | 0.0036 | 0.0037 | 0.0047 |
| semi-supervised (InfoMax vs GeMax) | - | 0.0005 | 0.0000 | 0.0067 | 0.0200 |

VGAE Kipf & Welling (2016b), all the models are trained on an incomplete version of these datasets where parts of the edges have been removed, while all node features are kept. We split the nodes of each dataset into three parts: 80% as training set, 10% as validation set and 10% as test set. We set $\gamma$ larger to emphasize the orthonormal representations regularization. The hyperparameters of Algorithm 1 are $\mu^0 = 0.5, \gamma = 5, \eta = 0.5, \epsilon = 0.3 \log n, \tau = 0.01$. The results in Table 4 show that GeMax methods outperform baselines.

Table 4: Performance of edge prediction. AUC is the area under ROC curve. AP is average precision.

| methods | Cora | | Citeseer | | Pubmed | |
|---|---|---|---|---|---|---|
| | AUC | AP | AUC | AP | AUC | AP |
| VGAE | 91.4±0.16 | 92.6±0.10 | 90.8±0.07 | 92.3±0.11 | 94.5±0.13 | 94.8±0.09 |
| ARGA | 92.3±0.21 | 92.5±0.13 | 93.1±0.16 | 93.5±0.25 | 96.3±0.26 | 96.8±0.15 |
| LGAE | 93.0±0.19 | 93.2±0.08 | 94.5±0.19 | 94.7±0.12 | 96.7±0.15 | 97.0±0.18 |
| GIC | 92.6±0.09 | 92.7±0.17 | 94.3±0.23 | 94.4±0.14 | 96.5±0.17 | 96.7±0.21 |
| InfoGraph-GeMax | 92.8±0.13 | 93.0±0.25 | 93.7±0.12 | 94.2±0.17 | 95.2±0.24 | 95.4±0.12 |
| GraphCL-GeMax | **93.2±0.24** | **93.4±0.19** | 94.6±0.20 | 94.9±0.23 | 96.6±0.13 | 96.7±0.10 |
| AutoGCL-GeMax | 93.1±0.15 | 93.2±0.21 | **94.9±0.14** | **95.1±0.13** | **96.9±0.19** | **97.2±0.15** |

## 5.3 MORE NUMERICAL RESULTS

The results of **parameter sensitivity analysis**, **ablation study**, **convergence analysis** and **time cost** are in the supplementary material.

## 6 CONCLUSIONS

In this work, we propose a novel Graph entropy Maximization (GeMax) method to learn the orthonormal representations, which can capture the most information of a graph. We also approximate the graph entropy via Shannon entropy and chromatic entropy. The experiment on unsupervised graph-level and node-level demenstrate the effectiveness of our GeMax method.

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

# A  APPENDIX

## A.1  NOTATIONS

The main notations used in this paper are shown in Table 5.

Table 5: Notations

| Symbol | Description | Symbol | Description |
|---|---|---|---|
| $G$ | a graph | $V$ | vertex set of graph $G$ |
| $\mathbf{g}$ | graph-level representation of $G$ | $n$ | the number of vertices of $G$ |
| $\boldsymbol{Z}$ | node representations matrix of $G$ | $\boldsymbol{z}_i$ | representation of node $i$ |
| $(G, P)$ | a probabilistic graph | $P(\mathbf{g}, \boldsymbol{Z})$ | probability distribution on $V$ |
| $\mathrm{VP}(G)$ | vertex packing polytope of $G$ | $P_i(\mathbf{g}, \boldsymbol{Z})$ | probability density of vertex $i$ |
| $H_k(G, P)$ | graph entropy | $H(P)$ | Shannon entropy |
| $H_c(G, P, \boldsymbol{\pi})$ | the entropy of a coloring $\boldsymbol{\pi}$ | $H_\chi(G, P)$ | chromatic entropy |
| $\boldsymbol{\pi}$ | a coloring on vertex set $V$ | $\boldsymbol{\Pi}(G)$ | set of all coloring $\boldsymbol{\pi}$ of $G$ |
| $\mathbb{C}_k$ | the set of vertices with color $k$ | $\alpha(G)$ | the independence number of $G$ |
| $\chi(G)$ | the chromatic number of $G$ | $D(G, P)$ | gap between bounds of $H_k(G, P)$ |
| $\theta$ | the parameters for learning $\mathbf{g}$ | $\phi$ | the parameters for learning $\boldsymbol{Z}$ |
| $\psi$ | the parameters for learning $\boldsymbol{\pi}$ | $\delta$ | error bound of the approximation |

## A.2  ERROR BOUND ANALYSIS OF THE APPROXIMATION

**Corollary A.1.** *Let* $\epsilon = \frac{\log \chi(G)}{1+\delta} + \log \alpha(G)$, *if* $H(P) \geq \epsilon$, *we have* $\frac{|H_k(G,P) - \hat{H}_k(G,P)|}{H_k(G,P)} \leq \delta$.

*Proof.* Let $D(G, P)$ be the gap between lower and upper bounds in Theorem 2.8, we have

$$D(G, P) := H_\chi(G, P) - H(P) + \log \alpha(G). \tag{24}$$

Let $\chi(G)$ be chromatic number of graph $G$ which is the smallest number of colors needed to color the vertices. The gap $D(G, P)$ is bounded by $\chi(G)$, $\alpha(G)$ and Shannon entropy $H(P)$ as follows.

**Corollary A.2** (Cardinal et al. (2004; 2005))**.**

$$0 \leq D(G, P) \leq \log \chi(G) + \log \alpha(G) - H(P).$$

Let $\hat{\mu}$ be an estimator of $\mu$ and $\hat{H}_k(G, P)$ be an approximation of $H_k(G, P)$ with respect to $\hat{\mu}$. Suppose $H(P) - \log \alpha(G) > 0$, the error bound of the approximation is as follows

$$\frac{|H_k(G, P) - \hat{H}_k(G, P)|}{H_k(G, P)} \leq \frac{D(G, P)}{H(P) - \log \alpha(G)} \leq \frac{\log \chi(G)}{H(P) - \log \alpha(G)} - 1. \tag{25}$$

Since $\alpha(G)$ and $\chi(G)$ are constants of a given $G$, maximizing the Shannon entropy $H(P)$ is to minimize the upper bound of the error bound in equation 25 such that it yields a more exact approximation. Let $\delta$ be the upper bound of the error bound of our approximation, we have

$$\frac{\log \chi(G)}{H(P) - \log \alpha(G)} - 1 \leq \delta \Rightarrow H(P) \geq \frac{\log \chi(G)}{1+\delta} + \log \alpha(G) \tag{26}$$

Thus, let $\epsilon = \frac{\log \chi(G)}{1+\delta} + \log \alpha(G)$, if $H(P) \geq \epsilon$, we have $\frac{|H_k(G,P) - \hat{H}_k(G,P)|}{H_k(G,P)} \leq \delta$. $\qquad\square$

## A.3  ARCHITECTURE AND GENERALIZATION

In this section, we introduce the architecture of our GeMax method is in Figure 2. In Figure 3, we apply GeMax on the architecture of InfoGraph by replacing the InfoMax loss with our GeMax objective $J(\mathcal{G}; \boldsymbol{\theta}, \boldsymbol{\phi}, \boldsymbol{\psi})$. In Figure 4, we show the architecture of Infograph Sun et al. (2019).

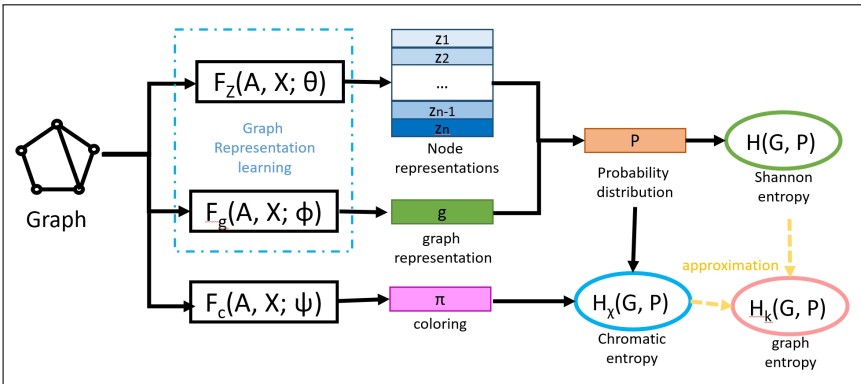

Figure 2: Architecture of GeMax

The graph representations learning part ($F_Z(\boldsymbol{A}, \boldsymbol{X}; \boldsymbol{\theta})$ and $F_g(\boldsymbol{A}, \boldsymbol{X}; \boldsymbol{\phi})$) in the blue dash box are not confined to any specific GNN models. Thus, GeMax can be applied on various GNN based models like InfoGraph Sun et al. (2019) or GraphCL You et al. (2020).

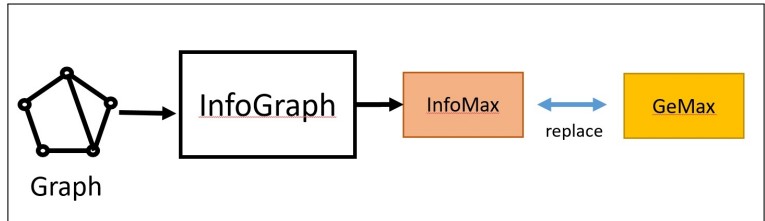

Figure 3: Applying GeMax to InfoGraph network by replacing the InfoMax loss

Given an unsupervised graph representation learning models, GeMax method is to replace the original loss with our GeMax objective $J(\mathcal{G}; \boldsymbol{\theta}, \boldsymbol{\phi}, \boldsymbol{\psi})$ equation 20 and introduce a coloring function $F_c(\boldsymbol{A}, \boldsymbol{X}; \boldsymbol{\psi})$. The $J(\mathcal{G}; \boldsymbol{\theta}, \boldsymbol{\phi}, \boldsymbol{\psi})$ can be optimized by our iterative algorithm.

### A.4 EXPERIMENT: STATISTICS OF DATASET

In this section, we provide the statistics of the dataset we used in experiments. For graph-level representation learning tasks, we use the TUdataset Morris et al. (2020) in Table 6. For node-level representation learning tasks, we use the network dataset Sen et al. (2008) in Table 7 for edge prediction. The TUdataset used in graph-level

Table 6: Statistics of TUdataset Morris et al. (2020)

| Name | Num. of graphs | Num. of classes | Num. of nodes | node labels | node attributes |
|---|---|---|---|---|---|
| MUTAG | 188 | 2 | 17.9 | yes | no |
| PROTEINS | 1113 | 2 | 39.1 | yes | yes |
| DD | 1178 | 2 | 284.32 | yes | no |
| NCI1 | 4110 | 2 | 29.9 | yes | no |
| COLLAB | 5000 | 3 | 74.49 | no | no |
| IMDB-B | 1000 | 2 | 19.8 | no | no |
| REDDIT-B | 2000 | 2 | 429.63 | no | no |
| REDDIT-M5K | 4999 | 5 | 508.52 | no | no |
| GITHUB | 12725 | 2 | 113.79 | no | no |

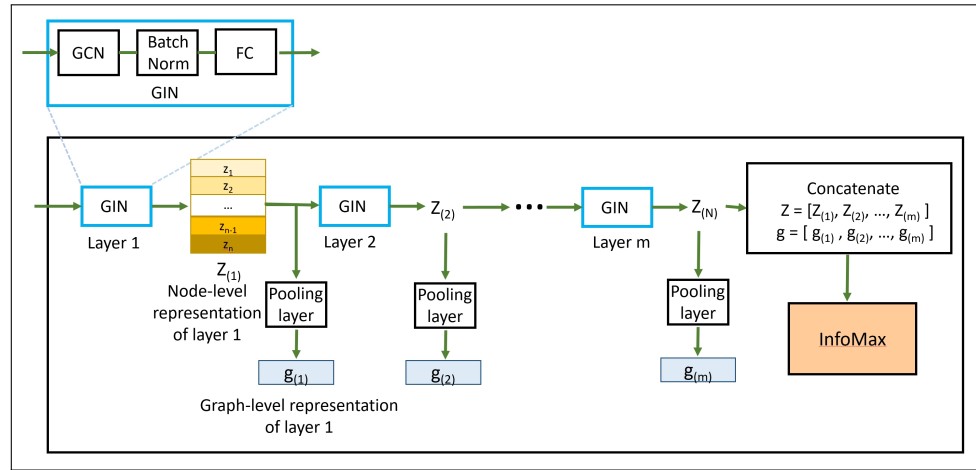

Figure 4: Architecture of InfoGraph with $m$ layers

Table 7: Statistics of Network dataset Sen et al. (2008) for edge prediction

| Name | Num. of nodes | Num. of node class | Num. of edges | described by a 0/1-valued |
|---|---|---|---|---|
| Cora | 2708 | 7 | 5429 | yes |
| Citeseer | 3312 | 6 | 4732 | yes |
| Pubmed | 19717 | 3 | 44338 | yes |

A.5 EXPERIMENT BASELINE: INFORMAX METHODS

For unsupervised and semi-supervised graph-level learning, those InfoMax based methods are the most current and influential methods spanning from 2019 to 2022, each boasting high citations on Google Scholar (see Table 8). Besides the InfoMax principle, there are few works on graph information bottleneck (GIB) Wu et al. (2020) and the subgraph information bottleneck (SIB) Yu et al. (2021). GIB and SIB aim to learn the minimal sufficient representation for downstream tasks. But GIB Wu et al. (2020) and SIB Yu et al. (2021) may fail if the downstream tasks are not available in the representation learning stage. Thus they are not suitable for unsupervised and semi-supervised graph learning such that they are not included in our baselines. To the best of our knowledge, we don't find other principles for unsupervised graph-level representation learning except InfoMax, GIB and SIB. Since the InfoMax methods are the most influential methods, we compare with them by replacing the InfoMax objective with our GeMax objective in equation 20 (see Figure 2 and Figure 3). In this work, we compare with five InfoMax based methods, that is, InfoGraph (Sun et al., 2019), GraphCL (You et al., 2020), AD-GCL (Suresh et al., 2021), JOAO (You et al., 2021) , AutoGCL (Yin et al., 2022). All these five methods share the same graph representation learning architecture with InfoGraph Sun et al. (2019), as shown in Figure 4.

Table 8: Google scholar citations comparison

| principle | InfoMax | | | | | GIB | SIB |
|---|---|---|---|---|---|---|---|
| methods | InfoGraph | GraphCL | AD-GCL | JOAO | AutoGCL | | |
| citations | 665 | 1101 | 176 | 249 | 42 | 129 | 26 |

Following (Nowozin et al., 2016; Sun et al., 2019; Belghazi et al., 2018), suppose the node-level representation $z_p(x)$ and the graph-level representation $g(x)$ are depending on the input $x$, $T_\varphi$ is a discriminator parameterized by a neural network with parameters $\varphi$, the Jensen-Shannon mutual information (MI) estimator (Fuglede & Topsoe, 2004; Nowozin et al., 2016; Hjelm et al., 2019; Sun

et al., 2019) $I_\varphi$ between $z_v$ and $g$ is defined as

$$I_\varphi(z_p, g) = \mathbb{E}_{\mathbb{P}}[-\text{sp}(-T_\varphi(z_p(x), g(x)))] - \mathbb{E}_{\mathbb{P} \times \tilde{\mathbb{P}}}[\text{sp}(T_\varphi(z_p(x'), g(x)))], \tag{27}$$

where $x$ is the input sample from distribution $\mathbb{P}$, $x'$ is the negative sample from distribution $\tilde{\mathbb{P}}$, and $\text{sp}(a) = \log(1 + e^a)$ denotes the softplus function. $\mathbb{P}$ is the empirical probability distribution of the input space and $\tilde{\mathbb{P}}$ is the empirical probability distribution of the negative input space. Many recent graph-level representation learning methods (Sun et al., 2019; You et al., 2020; Yin et al., 2022) are based on the InfoMax principle, i.e., maximizing equation 27. For example, InfoGraphSun et al. (2019) obtains graph-level representations by maximizing the mutual information between the graph-level representation and the node-level representations as follows

$$\phi^*, \theta^*, \varphi^* = \arg\max_{\phi, \theta, \varphi} \sum_{i=1}^{|\mathcal{G}|} \frac{1}{|V_i|} \sum_{p \in V_i} I_\varphi(z_p^\theta, g_i^\phi), \tag{28}$$

where $I_\varphi$ is the Jensen-Shannon MI estimator defined by equation 27. For semi-supervised learning, the dataset $\mathcal{G}$ is split into labeled dataset $\mathcal{G}^L$ and unlabeled dataset $\mathcal{G}^U$. They deploy another supervised encoder with parameter $\psi$ and then generate the supervised node-level representations $Z_i^\psi$, graph-level representations $g_i^\psi$ and prediction $\hat{y}_i^\psi$. The loss function of InfoGraph for semi-supervised learning is defined as follows:

$$\mathcal{L}_{\text{info-semi}} = \sum_{l=1}^{|\mathcal{G}^L|} \mathcal{L}_{\text{supervised}}(\hat{y}_l^\psi, y_l) + \sum_{i=1}^{|\mathcal{G}|} \mathcal{L}_{\text{unsupervised}}(Z_i^\theta, g_i^\phi) - \lambda \sum_{i=1}^{|\mathcal{G}|} \frac{1}{|V_i|} I_\varphi(g_i^\phi, g_i^\psi) \tag{29}$$

where $\mathcal{L}_{\text{unsupervised}}$ is derived from equation 28. The last term encourages the representations learned by the two encoders to have high mutual information.

## A.6 EXPERIMENT: GRAPH-LEVEL TASKS

**Unsupervised learning** Following (Sun et al., 2019; You et al., 2021; Yin et al., 2022), we train a graph representation model on unlabeled data to obtain graph representations and use these representations and graph labels to train a SVM classifier. Our experimental setup follows AutoGCL (Yin et al., 2022). Actually, all the four InfoMax methods (GraphCL, AD-GCL, JOJOv2 and AutoGCL) are based on the architecture of InfoGraph. Specifically, they use a 5-layer GIN Xu et al. (2018) with hidden size 128 as the representation model, shown in Figure 4. The model is trained with a batch size of 128 and a learning rate of 0.001. For those contrastive learning methods (e.g., JOJOv2 and AutoGCL), they use 30 epochs of contrastive pre-training under the naive strategy. We repeat the experiments for 10 times with different random seeds. In each time, we perform 10-fold cross-validation on each dataset. Specifically, in each fold, we use 90% of the total data as unlabeled data for contrastive pre-training and 10% as labeled testing data. The hyperparameters of Algorithm 1 are $\mu^0 = 0.5, \gamma = 0.5, \eta = 0.5, \epsilon = 0.3 \log n, \tau = 0.01$. We also conduct sensitivity analysis in Appendix A.8 to study how different hyperparameters affect the results. The average accuracy (ACC) and standard deviation are reported in Table 14.

**Semi-supervised Learning** Following (Hu et al., 2019; You et al., 2021; Yin et al., 2022), we compare our GeMax methods with InfoMax-based methods in semi-supervised learning tasks. The semi-supervised losses of InfoMax based methods were shown in equation 29 of Appendix A.5. To ensure fair comparisons, we follow the semi-supervised learning of InfoMax methods in equation 29 and replace the InfoMax objective with our GeMax objective in equation 20. Following the settings of AutoGCL (Yin et al., 2022), we employ a 10-fold cross-validation on each dataset. For each fold, we use 80% of the total data as the unlabeled data, 10% as labeled training data, and 10% as labeled testing data. The classifier for labeled data is a ResGCN (Chen et al., 2019) with 5 layers and a hidden size of 128. We repeat each experiment 10 times and report the average accuracy in Table 2.

## A.7 EXPERIMENT: NODE-LEVEL TASKS

Though original motivation of orthonormal representations in equation 1 is from information theory, it can be used to reconstruct the adjacency matrix $A$ using the information of non-adjacency.

Denoting $\hat{A}$ as the reconstructed adjacency matrix, we have

$$\hat{A} = \sigma(\mathbf{abs}(\boldsymbol{Z}^{\phi}(\boldsymbol{Z}^{\phi})^{\top} - \boldsymbol{I}_n)), \tag{30}$$

where the $\mathbf{abs}(\cdot)$ is the element-wise absolute value function and the $\sigma(\cdot)$ is a element-wise sigmoid function. Thus, we can compare our GeMax methods with other graph reconstruction methods such as VGAE Kipf & Welling (2016b), ARGA Pan et al. (2018), GIC Mavromatis & Karypis (2020) and LGAE Salha et al. (2021). We use the network architecture of InfoMax methods (InfoGraph, GraphCL and AutoGCL) by replacing the InfoMax objective with our GeMax objective. Since the orthonormal representations mainly contributes to graph reconstruction and edge prediction, we should set the parameter of orthonormal representations regularization (i.e., $\eta$) larger. The hyperparameters of Algorithm 1 are $\mu^0 = 0.5, \gamma = 0.5, \eta = 5, \epsilon = 0.3 \log n, \tau = 0.01$.

Following VGAE Kipf & Welling (2016b), all the models are trained on an incomplete version of these datasets where parts of the edges have been removed, while all node features are kept. We split the nodes of each dataset into three parts: 80% as training set, 10% as validation set and 10% as test set. We report area under the ROC curve (AUC) and average precision (AP) scores for each model on the test set in Table 4.

### A.8 EXPERIMENT: SENSITIVITY ANALYSIS OF HYPERPARAMETERS

In the alternative algorithm 1, there are four hyperparameters need to be tuned: the initial approximation weight $\mu^0$, the orthonormal representation regularization parameter $\gamma$, the coloring regularization parameter $\eta$, the lower threshold of average Shannon entropy $\epsilon$. In this section we analyse the parameter sensitivity on the InfoGraph Sun et al. (2019) with different hyperparameters. We repeat each experiments for ten times and plot the average accuracy with variance on different datasets.

### A.8.1 THE INITIAL APPROXIMATION WEIGHT

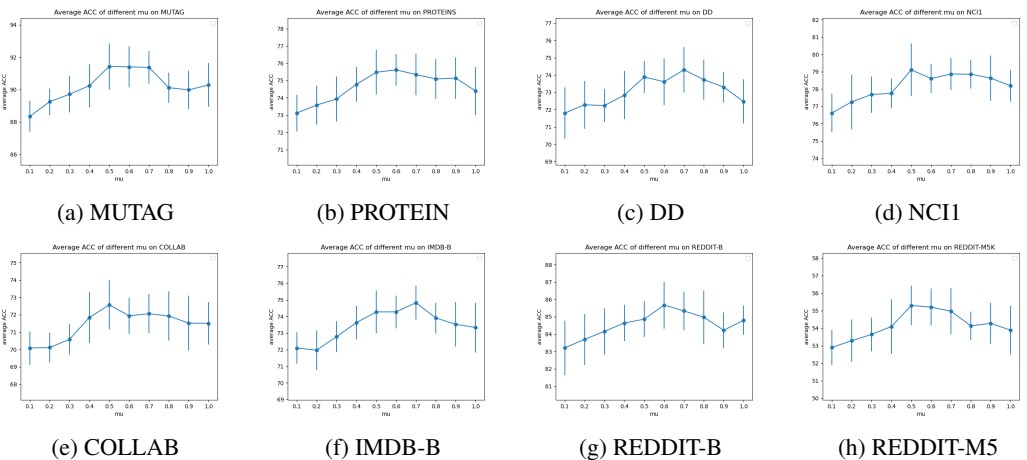

(a) MUTAG      (b) PROTEIN      (c) DD      (d) NCI1

(e) COLLAB      (f) IMDB-B      (g) REDDIT-B      (h) REDDIT-M5

Figure 5: The average ACC of different $\mu$ on different data

The approximation weight $\mu$ is initialized as $\mu = \mu^0$ in the beginning of algorithm 1. In Figure 5, we fix $\epsilon = 0.3 \log n$ and other hyperparameters. We tune $\mu^0$ from $\{0.1, 0.2, ..., 0.9, 1\}$. The results show that algorithm 1 achieves the top performance when $0.5 \leq \mu^0 \leq 0.7$. If $\mu^0 = 0$, the approximation to the graph entropy starts with $\hat{H}_k(G, P) = H_c(G_j, P(\mathbf{g}_j^{\theta^0}, \boldsymbol{Z}_j^{\phi^0}), \boldsymbol{\pi}^{\psi^0})$. However, a very small $\mu^0$ adversely affect the performance because the coloring $\boldsymbol{\pi}^{\psi^0}$ is randomly initialized in the beginning. If $\mu^0 = 1$, the algorithm 1 degenerates to equation 11, that is, maximizing the lower bound of graph entropy. If the equality in Corollary 2.9 holds, $\mu^0 = 1$ can result in learning the representations for exact graph entropy $H_k(G, P)$. However, if the equality in Corollary 2.9 doesn't hold, the approximation will be inexact and thus the performance decreases.

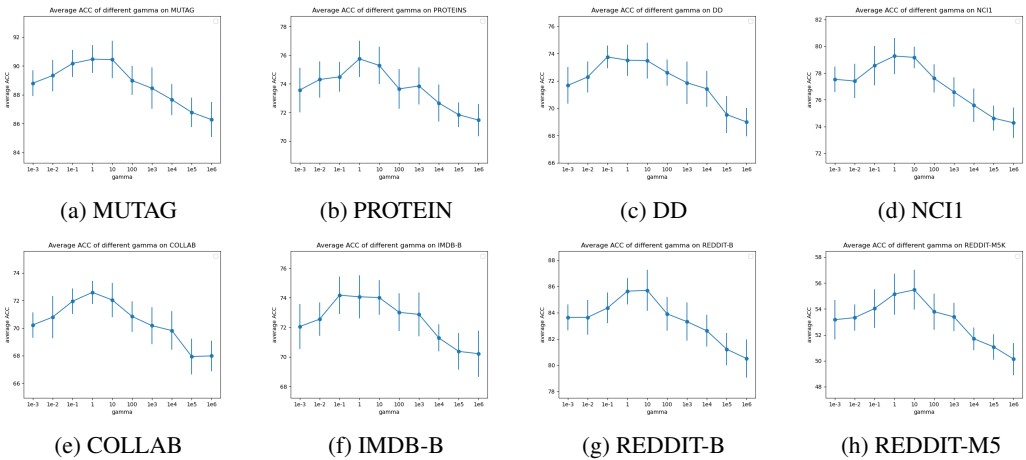

Figure 6: The average ACC of different $\gamma$ on different data

### A.8.2 THE ORTHONORMAL REPRESENTATION REGULARIZATION PARAMETER

$\gamma$ is the hyperparameter for orthonormal representation regularization. In Figure 6, we fix other hyperparameters and tune $\gamma$ from $\{10^{-3}, 10^{-2}, ..., 10^5, 10^6\}$. The results show that $\gamma$ is not sensitive when $1 \leq \gamma \leq 10$. If $\gamma$ is too small, the performance decreases because the node-level representations $Z$ may not be orthonormal representations. A very large $\gamma$ adversely affect the performance because the orthonormal representation regularization dominates the representation learning.

### A.8.3 THE COLORING REGULARIZATION PARAMETER

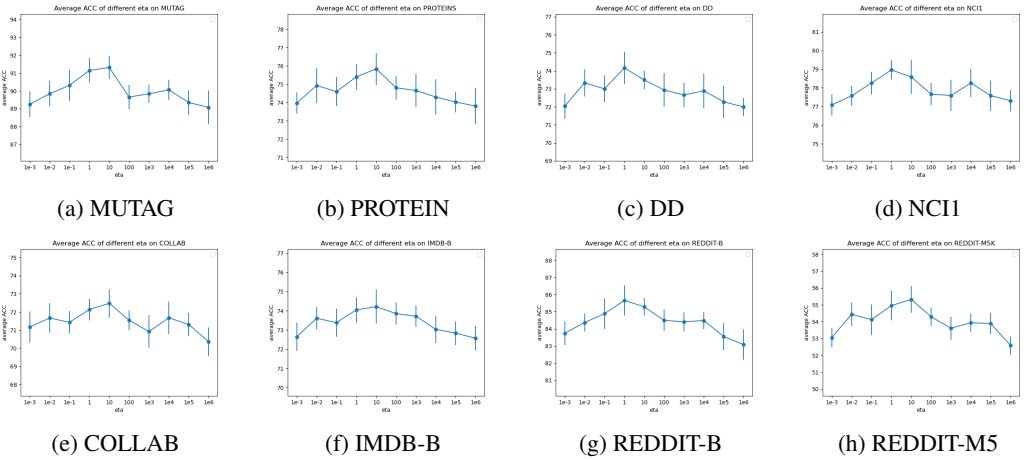

Figure 7: The average ACC of different $\eta$ on different data

$\eta$ is the hyperparameter for coloring regularization. In Figure 7, we fix other hyperparameters and tune $\eta$ from $\{10^{-3}, 10^{-2}, ..., 10^5, 10^6\}$. The results show that algorithm 1 achieve top comference when $1 \leq \eta \leq 10$. If $\eta$ is too small, the performance decreases because the the coloring function $F_c$ are unable to search for a coloring for $G$. A very large $\eta$ adversely affect the performance because the coloring searched by $F_c$ is not related to the chromatic entropy.

### A.8.4 THE LOWER THRESHOLD OF AVERAGE SHANNON ENTROPY

In algorithm 1, we use $\epsilon$ to control the updating of $\mu$. In Figure 8, we fix other hyperparameters and tune $\epsilon$ ranging from $0.1 \log n$ to $\log n$. The results show that $\epsilon$ is not sensitive when $0.3 \leq \epsilon \leq 0.4$.

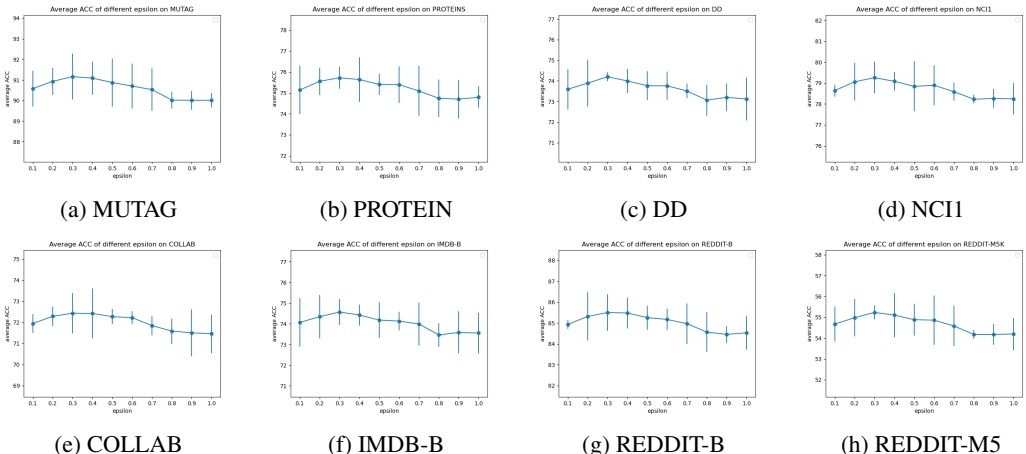

Figure 8: The average ACC of different $\epsilon$ on different data

If $\epsilon$ is too small, the algorithm 1 degenerates to equation 11, that is, maximizing the lower bound of graph entropy. If the equality in Corollary 2.9 holds, $\mu^0 = 1$ can result in learning the representations for exact graph entropy $H_k(G, P)$. However, if the equality in Corollary 2.9 doesn't hold, the approximation will be inexact and thus the performance decreases. If $\epsilon$ is large, the $\mu$ will not be updated such that the error bound are not guaranteed to be smaller than $\delta$. Thus the performance decreases.

## A.9 EXPERIMENT: ABLATION STUDY

In the ablation study, we analyse the importance of each part of GeMax objective $J(\mathcal{G}; \boldsymbol{\theta}, \boldsymbol{\phi}, \boldsymbol{\psi})$.

### A.9.1 REMOVE THE ORTHONORMAL REPRESENTATION REGULARIZATION

We remove the orthonormal representation regularization of GeMax objective $J(\mathcal{G}; \boldsymbol{\theta}, \boldsymbol{\phi}, \boldsymbol{\psi})$ in equation 20 by setting $\gamma = 0$. The results in Table 9 show that the orthonormal representation regularization can improve the performance of graph representation learning.

Table 9: Performance (ACC) of unsupervised learning for Ablation study. The ablation indicates $\gamma = 0$ in equation 20. The **bold** numbers denote the better performances of the same method.

| methods | ablation | MUTAG | PROTEINS | DD | NCI1 | COLLAB | IMDB-B | REDDIT-B | REDDIT-M5K |
|---------|----------|-------|----------|-----|------|--------|--------|----------|------------|
| InfoGraph | ✓ | 87.48±1.56 | 73.29±1.13 | 72.54±1.27 | 76.89±1.28 | 70.22±1.97 | 71.53±1.31 | 84.07±1.16 | 54.32±1.53 |
|  | × | **91.13±1.70** | **75.77±1.26** | **74.16±1.65** | **79.24±1.43** | **72.57±1.74** | **74.59±1.53** | **85.53±1.92** | **55.21±1.69** |
| GraphCL | ✓ | 88.39±1.12 | 74.43±1.28 | 76.77±1.22 | 76.09±1.26 | 70.32±1.85 | 71.07±1.36 | 88.73±1.52 | 54.69±1.49 |
|  | × | **90.36±1.69** | **76.86±1.62** | **79.25±1.53** | **78.72±1.79** | **73.43±1.62** | **73.12±1.25** | **91.47±1.74** | **56.25±1.53** |
| AD-GCL | ✓ | 87.45±1.31 | 73.01±1.55 | 76.23±1.86 | 74.73±1.19 | 72.85±1.37 | 71.64±1.13 | 86.65±1.28 | 54.91±1.59 |
|  | × | **89.68±1.47** | **74.52±1.71** | **77.58±1.41** | **76.35±1.62** | **74.83±1.79** | **73.52±1.45** | **88.03±1.62** | **55.03±1.54** |
| JOAOv2 | ✓ | 87.69±1.81 | 73.05±1.16 | 70.03±1.83 | 72.67±1.98 | 70.11±1.50 | 71.37±1.56 | 80.24±1.35 | 55.42±1.33 |
|  | × | **88.33±1.58** | **74.63±1.87** | **72.60±1.35** | **75.36±1.42** | **71.68±1.67** | **72.21±1.72** | **81.68±1.40** | **57.17±1.67** |
| AutoGCL | ✓ | 87.23±1.48 | 74.36±1.65 | 76.79±1.60 | 81.76±1.55 | 70.48±1.08 | 72.63±1.54 | 87.03±1.95 | 54.27±1.61 |
|  | × | **90.85±1.28** | **76.23±1.29** | **78.36±1.51** | **83.21±1.34** | **72.39±1.57** | **74.05±1.79** | **90.42±1.31** | **56.81±1.85** |

### A.9.2 REMOVE THE GRAPH ENTROPY

We remove the graph entropy of GeMax objective $J(\mathcal{G}; \boldsymbol{\theta}, \boldsymbol{\phi}, \boldsymbol{\psi})$ in equation 20 by setting $\gamma = 10^5$ such that the orthonormal representation will dominate the optimization. The results in Table 10 show that the graph entropy can improve the performance of graph representation learning.

### A.9.3 REMOVE THE CHROMATIC ENTROPY

We remove the chromatic entropy of GeMax objective $J(\mathcal{G}; \boldsymbol{\theta}, \boldsymbol{\phi}, \boldsymbol{\psi})$ in equation 20 by setting $\mu = 1$. The results in Table 11 show that the chromatic entropy can improve the performance of graph representation learning.

Table 10: Performance (ACC) of unsupervised learning for Ablation study. The ablation indicates $\gamma = 10^5$ in equation 20. The **bold** numbers denote the better performances of the same method.

| methods | ablation | MUTAG | PROTEINS | DD | NCI1 | COLLAB | IMDB-B | REDDIT-B | REDDIT-M5K |
|---|---|---|---|---|---|---|---|---|---|
| InfoGraph | ✓ | 82.53±1.24 | 70.57±1.42 | 71.18±1.34 | 73.26±1.45 | 68.73±1.43 | 70.11±1.09 | 82.43±1.24 | 51.53±1.42 |
| | × | **91.13±1.70** | **75.77±1.26** | **74.16±1.65** | **79.24±1.43** | **72.57±1.74** | **74.59±1.53** | **85.53±1.92** | **55.21±1.69** |
| GraphCL | ✓ | 83.54±1.47 | 71.39±1.65 | 73.26±1.03 | 75.86±1.31 | 69.07±1.37 | 70.22±1.87 | 85.98±1.47 | 52.37±1.50 |
| | × | **90.36±1.69** | **76.86±1.62** | **79.25±1.53** | **78.72±1.79** | **73.43±1.62** | **73.12±1.25** | **91.47±1.74** | **56.25±1.53** |
| AD-GCL | ✓ | 85.49±1.84 | 70.16±1.36 | 74.49±1.78 | 71.05±1.32 | 70.58±1.73 | 70.90±1.65 | 83.47±1.67 | 52.64±1.83 |
| | × | **89.68±1.47** | **74.52±1.71** | **77.58±1.41** | **76.35±1.62** | **74.83±1.79** | **73.52±1.45** | **88.03±1.62** | **55.03±1.54** |
| JOAOv2 | ✓ | 82.73±1.69 | 70.48±1.43 | 68.24±1.69 | 70.04±1.25 | 69.27±1.47 | 70.63±1.04 | 78.15±1.57 | 52.79±1.27 |
| | × | **88.33±1.58** | **74.63±1.87** | **72.60±1.35** | **75.36±1.42** | **71.68±1.67** | **72.21±1.72** | **81.68±1.40** | **57.17±1.67** |
| AutoGCL | ✓ | 83.14±1.58 | 71.93±1.82 | 74.35±1.51 | 79.11±1.57 | 68.26±1.83 | 71.48±1.57 | 82.83±1.72 | 53.07±1.63 |
| | × | **90.85±1.28** | **76.23±1.29** | **78.36±1.51** | **83.21±1.34** | **72.39±1.57** | **74.05±1.79** | **90.42±1.31** | **56.81±1.85** |

Table 11: Performance (ACC) of unsupervised learning for Ablation study. The ablation indicates $\mu = 1$ in equation 20. The **bold** numbers denote the better performances of the same method.

| methods | ablation | MUTAG | PROTEINS | DD | NCI1 | COLLAB | IMDB-B | REDDIT-B | REDDIT-M5K |
|---|---|---|---|---|---|---|---|---|---|
| InfoGraph | ✓ | 88.52±1.49 | 74.16±1.37 | 72.98±1.52 | 78.30±1.67 | 71.95±1.43 | 72.46±1.08 | 84.91±1.34 | 54.96±1.53 |
| | × | **91.13±1.70** | **75.77±1.26** | **74.16±1.65** | **79.24±1.43** | **72.57±1.74** | **74.59±1.53** | **85.53±1.92** | **55.21±1.69** |
| GraphCL | ✓ | 89.27±1.54 | 75.16±1.43 | 77.73±1.28 | 77.19±1.69 | 72.21±1.54 | 72.21±1.21 | 90.34±1.86 | 55.10±1.88 |
| | × | **90.36±1.69** | **76.86±1.62** | **79.25±1.53** | **78.72±1.79** | **73.43±1.62** | **73.12±1.25** | **91.47±1.74** | **56.25±1.53** |
| AD-GCL | ✓ | 88.98±1.06 | 73.89±1.38 | 76.95±1.53 | 75.27±1.93 | 73.04±1.53 | 72.45±1.22 | 87.11±1.90 | 54.71±1.34 |
| | × | **89.68±1.47** | **74.52±1.71** | **77.58±1.41** | **76.35±1.62** | **74.83±1.79** | **73.52±1.45** | **88.03±1.62** | **55.03±1.54** |
| JOAOv2 | ✓ | 87.52±1.69 | 73.79±1.93 | 71.52±1.58 | 74.70±1.31 | 71.19±1.74 | 72.08±1.38 | 80.49±1.05 | 56.23±1.78 |
| | × | **88.33±1.58** | **74.63±1.87** | **72.60±1.35** | **75.36±1.42** | **71.68±1.67** | **72.21±1.72** | **81.68±1.40** | **57.17±1.67** |
| AutoGCL | ✓ | 88.43±1.62 | 75.03±1.71 | 76.72±1.89 | 82.89±1.47 | 71.97±1.72 | 73.21±1.43 | 89.14±1.76 | 54.86±1.49 |
| | × | **90.85±1.28** | **76.23±1.29** | **78.36±1.51** | **83.21±1.34** | **72.39±1.57** | **74.05±1.79** | **90.42±1.31** | **56.81±1.85** |

### A.9.4 REMOVE THE SHANNON ENTROPY

We remove the Shannon entropy of GeMax objective $J(\mathcal{G}; \boldsymbol{\theta}, \boldsymbol{\phi}, \boldsymbol{\psi})$ in equation 20 by setting $\mu = 0$. The results in Table 12 show that the Shannon entropy can improve the performance of graph representation learning.

Table 12: Performance (ACC) of unsupervised learning for Ablation study. The ablation indicates $\mu = 0$ in equation 20. The **bold** numbers denote the better performances of the same method.

| methods | ablation | MUTAG | PROTEINS | DD | NCI1 | COLLAB | IMDB-B | REDDIT-B | REDDIT-M5K |
|---|---|---|---|---|---|---|---|---|---|
| InfoGraph | ✓ | 87.26±1.79 | 74.64±1.71 | 71.18±1.43 | 75.96±1.57 | 71.45±1.53 | 72.01±1.86 | 84.25±1.87 | 54.68±1.93 |
| | × | **91.13±1.70** | **75.77±1.26** | **74.16±1.65** | **79.24±1.43** | **72.57±1.74** | **74.59±1.53** | **85.53±1.92** | **55.21±1.69** |
| GraphCL | ✓ | 88.90±1.30 | 73.93±1.75 | 77.65±1.34 | 76.28±1.49 | 71.82±1.44 | 71.69±1.20 | 87.16±1.35 | 53.78±1.51 |
| | × | **90.36±1.69** | **76.86±1.62** | **79.25±1.53** | **78.72±1.79** | **73.43±1.62** | **73.12±1.25** | **91.47±1.74** | **56.25±1.53** |
| AD-GCL | ✓ | 87.08±1.12 | 72.95±1.32 | 75.26±1.43 | 74.59±1.47 | 71.61±1.42 | 70.45±1.48 | 85.62±1.76 | 54.87±1.35 |
| | × | **89.68±1.47** | **74.52±1.71** | **77.58±1.41** | **76.35±1.62** | **74.83±1.79** | **73.52±1.45** | **88.03±1.62** | **55.03±1.54** |
| JOAOv2 | ✓ | 86.47±1.57 | 72.74±1.32 | 71.38±1.19 | 73.53±1.47 | 68.21±1.83 | 70.21±1.84 | 78.35±1.47 | 53.08±1.43 |
| | × | **88.33±1.58** | **74.63±1.87** | **72.60±1.35** | **75.36±1.42** | **71.68±1.67** | **72.21±1.72** | **81.68±1.40** | **57.17±1.67** |
| AutoGCL | ✓ | 87.57±1.76 | 73.26±1.48 | 74.49±1.81 | 80.22±1.59 | 70.46±1.93 | 72.35±1.17 | 86.49±1.67 | 53.57±1.90 |
| | × | **90.85±1.28** | **76.23±1.29** | **78.36±1.51** | **83.21±1.34** | **72.39±1.57** | **74.05±1.79** | **90.42±1.31** | **56.81±1.85** |

### A.10 EXPERIMENT: CONVERGENCE ANALYSIS

In Figure 9, we can see that the orthonormal representation loss $\ell_{\text{orth}}$ and the coloring loss $\ell_c$ decrease into a small value. The Shannon entropy $H(P)$ and the chromatic entropy $H_\chi(G, P)$ converge into a stable value. Thus, the overall objective $J(\mathcal{G}; \boldsymbol{\theta}, \boldsymbol{\phi}, \boldsymbol{\psi})$ converges.

### A.11 EXPERIMENT: EXACT PENALTY METHOD

We propose a exact penalty algorithm 2 to solve the problem 14 as follows. As $\gamma$ and $\eta$ increasing into a large value, the constraints will be satisfied. We repeat the unsupervised experiments using algorithm 2 and report the results in Table 13. we have found that the representations given the regularized optimization are as good as those given by the constrained optimization.

### A.12 EXPERIMENT: TIME COST

We run the programming on a machine with Intel 7 CPU and RTX 3090 GPU. We repeat the experiment for five times and report the results in Table **??**.

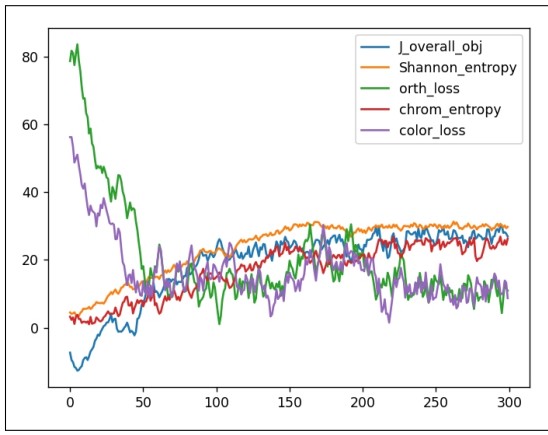

Figure 9: Convergence analysis of each part of the GeMax objective

---

**Algorithm 2** exact penalty method

1: **Initialization:** $\boldsymbol{\theta}^0, \boldsymbol{\phi}^0, \boldsymbol{\psi}^0, \mu^0 = 0.5, \gamma = 0.5, \eta = 0.5, \epsilon$ (e.g., $0.3 \log n$), $\varepsilon$ (e.g., $0.01$).
2: **repeat**
3:     $\theta^{t+1}, \phi^{t+1} = \operatorname{argmax}_{\boldsymbol{\theta}, \boldsymbol{\phi}} J(\mathcal{G}; \boldsymbol{\theta}, \boldsymbol{\phi}, \boldsymbol{\psi}^t)$
4:     $\boldsymbol{\psi}^{t+1} = \operatorname{argmin}_{\boldsymbol{\psi}} J(\mathcal{G}; \boldsymbol{\theta}^{t+1}, \boldsymbol{\phi}^{t+1}, \boldsymbol{\psi})$
5:     $\gamma^{t+1} = \gamma^t + 0.005, \eta^{t+1} = \eta^t + 0.005$
6:     **if** $\bar{H}(\mathcal{G}; t) < \epsilon$ and $\mu^t + 0.01 \leq 1$ **then** $\mu^{t+1} = \mu^t + 0.01$ **else** $\mu^{t+1} = \mu^t$
7: **until** $|J(\mathcal{G}; \boldsymbol{\theta}^{t+1}, \boldsymbol{\phi}^{t+1}, \boldsymbol{\psi}^{t+1}) - J(\mathcal{G}; \boldsymbol{\theta}^t, \boldsymbol{\phi}^t, \boldsymbol{\psi}^t)| \leq \varepsilon$

---

Table 13: Performance (ACC) of unsupervised learning. regularized opt. denotes the regularized algorithm 1 and constrained opt. denotes the exact algorithm 2.The **bold** numbers denote the better performances of the same method.

| methods | algorithm | MUTAG | PROTEINS | DD | NCI1 |
|---|---|---|---|---|---|
| InfoGraph | regularized opt. | 91.13±1.70 | **75.77±1.26** | 74.16±1.65 | **79.24±1.43** |
| | constrained opt. | **91.67±1.52** | 75.39±1.75 | **75.21±1.70** | 79.16±1.23 |
| GraphCL | regularized opt. | **90.36±1.69** | **76.86±1.62** | 79.25±1.53 | 78.72±1.79 |
| | constrained opt. | 90.05±1.87 | 76.04±1.91 | **79.63±1.67** | **78.95±1.83** |
| AD-GCL | regularized opt. | **89.68±1.47** | 74.52±1.71 | 77.58±1.41 | **76.35±1.62** |
| | constrained opt. | 89.25±1.52 | **74.71±1.53** | **77.87±1.22** | 75.73±1.29 |

Table 14: Time cost. The h is for hour and the m is for minute.

| tasks | methods and principles | | MUTAG | PROTEINS | DD | NCI1 | COLLAB | IMDB-B | REDDIT-B | REDDIT-M5K |
|---|---|---|---|---|---|---|---|---|---|---|
| unsupervised learning | InfoGraph | InfoMax | 2.2 m | 11.3 m | 1 h 32 m | 38.1 m | 1 h 36 m | 4.7 m | 3 h 16 m | 7 h 25 m |
| | | GeMax | 2.4 m | 12.7 m | 1 h 26 m | 37.2 m | 1 h 48 m | 5.2 m | 3 h 27 m | 7 h 47 m |
| | GraphCL | InfoMax | 3.2 m | 17.2 m | 1 h 54 m | 51.8 m | 2 h 23 m | 6.1 m | 4 h 49 m | 10 h 25 m |
| | | GeMax | 3.5 m | 16.8 m | 2 h 03 m | 58.3 m | 2 h 32 m | 7.5 m | 4 h 31 m | 10 h 46 m |
| | AD-GCL | InfoMax | 4.7 m | 26.4 m | 2 h 35 m | 1h 7 m | 2 h 48 m | 15.5 m | 5 h 37 m | 14 h 16 m |
| | | GeMax | 4.5 m | 25.1 m | 2 h 42 m | 1h 16 m | 2 h 57 m | 13.2 m | 5 h 26 m | 13 h 52 m |
| | JOAOv2 | InfoMax | 5.7 m | 33.8 m | 3 h 2 m | 1h 29 m | 3 h 10 m | 23.6 m | 6 h 7 m | 15 h 35 m |
| | | GeMax | 6.4 m | 35.2 m | 3 h 18 m | 1h 15 m | 3 h 23 m | 24.1 m | 6 h 15 m | 15 h 26 m |
| | AutoGCL | InfoMax | 6.8 m | 42.7 m | 3 h 27 m | 1h 56 m | 3 h 47 m | 32.4 m | 6 h 35 m | 16 h 43 m |
| | | GeMax | 6.3 m | 41.2 m | 3 h 34 m | 2h 7 m | 3 h 52 m | 34.1 m | 6 h 46 m | 16 h 56 m |
| semi-supervised learning | InfoGraph | InfoMax | - | 12.7 m | 2 h 8 m | 55.3 m | 2 h 38 m | - | 4 h 2 m | 9 h 17 m |
| | | GeMax | - | 13.9 m | 2 h 12 m | 57.2 m | 2 h 49 m | - | 4 h 13 m | 9 h 26 m |
| | GraphCL | InfoMax | - | 23.4 m | 2 h 45 m | 1 h 7 m | 2 h 42 m | - | 5 h 36 m | 12 h 7 m |
| | | GeMax | - | 25.3 m | 2 h 57 m | 1 h 13 m | 2 h 56 m | - | 5 h 43 m | 12 h 23 m |
| | AD-GCL | InfoMax | - | 35.1 m | 3 h 24 m | 1 h 39 m | 3 h 15 m | - | 6 h 26 m | 14 h 33 m |
| | | GeMax | - | 38.6 m | 3 h 17 m | 1 h 49 m | 3 h 21 m | - | 6 h 43 m | 14 h 47 m |
| | JOAOv2 | InfoMax | - | 43.8 m | 3 h 31 m | 2 h 1 m | 4 h 23 m | - | 7 h 12 m | 17 h 26 m |
| | | GeMax | - | 46.2 m | 3 h 49 m | 2 h 17 m | 4 h 18 m | - | 7 h 29 m | 17 h 49 m |
| | AutoGCL | InfoMax | - | 49.3 m | 3 h 40 m | 2 h 13 m | 4 h 28 m | - | 7 h 36 m | 18 h 56 m |
| | | GeMax | - | 53.4 m | 3 h 54 m | 2 h 27 m | 4 h 39 m | - | 7 h 41 m | 18 h 15 m |

