# OpenReview forum: "Learning Graph Representations via Graph Entropy Maximization"
_ICLR.cc/2024/Conference — Submitted to ICLR 2024_

### Official Review · Reviewer_Dct6 · 2023-10-22

**Soundness:** 2 fair
**Presentation:** 2 fair
**Contribution:** 2 fair
**Rating:** 3
**Confidence:** 3

**Summary:**

This paper proposes a measure of graph information by graph entropy. Since direct optimization of graph entropy is known to be NP hard, this paper provides an approximation of the graph entropy for the practical use of this entropy.
This paper also provides experimental results.

**Strengths:**

1) Formulation of the proposed objective function Eq.(14). This is also supported by some standard Shannon entropy theories (Thm4.1-Cor4.4)

**Weaknesses:**

1) The proposed method is almost off-the-shelf application of the existing results and methods. Maybe Eq.(9) is a proposal, which is a direct application of Def 3.1 to entropy. However, I speculate the rest of the discussion is off-the-shelf.


2) This might be related to 1). It is hard to distinguish which are the authors' results and which are the existing results. For the numbered theoretical claims, as far as I understand only Car 4.6 is the authors result. Then, why the others are not in the preliminaries? All the discussion between Eq. (6) and Eq.(23) is not the authors' own?

**Questions:**

1) While I appreciate the authors conduct experiments on many methods in Table 1 and Table 2, I am wondering if the authors have any other baseline of an entropy component other than InfoMax?

---

> ### Author Response · Authors · 2023-11-21
>
> **Weaknesses:**
>
> *Off-the-Shelf Application:* We appreciate the reviewer's feedback regarding the perceived off-the-shelf application of existing results and methods in our paper. To address this concern, we have restructured our paper to improve clarity. We have relocated all mathematical content related to other researchers' work to the preliminary section, ensuring a clear separation between our contributions and existing research.
>
> **Our contributions:**
>
> 1. Our primary contribution is the introduction of Graph Entropy Maximization (GeMax) for graph representation learning, marking the first instance of its application to the graph learning community. As emphasized in the introduction section, "graph entropy" has been used in graph learning, but it typically refers to structural entropy and differs from the genuine Graph entropy as defined by Körner (1973). We have highlighted that Graph entropy is a fundamental concept in combinatorics and information theory, with a rich history and contributions from mathematicians such as Claude E. Shannon and János Körner. When researchers create entropy-based methods for graph learning, it is essential to cite and introduce the important works of these mathematicians.
>
> 2. In the problem setup section, we present a framework for applying Graph entropy to graph learning problems, clearly defining our contributions in Problem Eq. (10). These contributions include the direct measurement of information within orthonormal representations, supported by Corollary 2.4, and the introduction of a Gaussian distribution with the vertex set, detailed in Eq. (8), to facilitate the application of Graph entropy.
>
> 3. In the methodology section, we introduce an approximation method for Graph entropy computation due to its NP-hard nature. This method utilizes Shannon entropy and chromatic entropy, resulting in a max-min optimization problem solved through alternating minimization. Additionally, we provide theoretical error bounds to assess the accuracy of our approximation.
>
> It is crucial to emphasize that equations without citations represent our original contributions, ensuring a clear distinction between our work and that of other researchers.
>
> **Questions:**
>
> *Baseline of Entropy Component:* We discuss other unsupervised graph representation learning principles in Appendix A.5. For unsupervised and semi-supervised graph-level learning, InfoMax-based methods are the most current and influential methods spanning from 2019 to 2022, each boasting high citations on Google Scholar (see Table 8). Besides the InfoMax principle, there are few works on graph information bottleneck (GIB) [1] and the subgraph information bottleneck (SIB)  [2]. GIB and SIB aim to learn the minimal sufficient representation for downstream tasks. But GIB [1] and SIB [2] may fail if the downstream tasks are not available in the representation learning stage. Thus, they are not suitable for unsupervised and semi-supervised graph learning, and that's why they are not included in our baselines. To the best of our knowledge, we don't find other principles for unsupervised graph-level representation learning except InfoMax, GIB, and SIB. Since the InfoMax methods are the most influential methods, we compare with them by replacing the InfoMax objective with our GeMax objective.
>
> | methods   | InfoGraph | GraphCL | AD-GCL | JOAO | AutoGCL | GIB | SIB |
> |-----------|-----------|---------|--------|------|---------|-----|-----|
> | citations | 665       | 1101    | 176    | 249  | 42      | 129 | 26  |
>
> [1] Tailin Wu, Hongyu Ren, Pan Li, and Jure Leskovec. Graph information bottleneck. NIPS, 2020
>
> [2]Junchi Yu, Tingyang Xu, Yu Rong, Yatao Bian, Junzhou Huang, and Ran He. Recognizing predictive
> substructures with subgraph information bottleneck. PAMI, 2021

---

### Official Review · Reviewer_zWFj · 2023-10-29

**Soundness:** 3 good
**Presentation:** 3 good
**Contribution:** 2 fair
**Rating:** 5
**Confidence:** 4

**Summary:**

This paper proposed a new approach to derive graph-level representation using graph entropy. Since the computation of graph entropy is a NP-hard problem, they provided thorough derivation to approximate the graph entropy through graph neural networks. Specifically, they derived the loss function to be the combination of Shannon entropy and chromatic entropy, on top of which they added orthonormal loss and coloring loss for regularization. Ablation study is done for these components for their impact.

**Strengths:**

• Provided a novel way to design graph level representation. The maximization of graph entropy aims to capture the most information from a given graph, focusing on the orthogonal vertices.
• Clearly outlined the derivation of the loss function of graph level entropy maximization. They also provided a clear summary of the notations in Appendix (table 5). It would be better if the author can include the shapes of dimensions of the variables.
• Good ablation study in terms of the components in the loss function.

**Weaknesses:**

• Compared to InfoGraph, this paper proposes to use 3 GNNs (graph representation, node representation and coloring representation), for the calculation of the loss function. This could be a problem in terms of model size and running time. The authors provided very limited comments and experiments on this matter.
• Despite more parameters introduced in the proposed method, the performance improvement on different datasets is generally small, especially on larger and more complex datasets.
• For time analysis in table 14, only the running time for InfoGraph is presented. What about other networks? More importantly, what about GeMax compared with InfoMax?
• Figure 2 and Figure 3 & 4 in appendix have many duplicated components. Figure 2 and Figure 3 are especially similar. The author should find a way to condense the information shown.
• The author seems to use the same citation for “InfoMax” and “InfoGraph”. Their method is comparable to “InfoMax” used in “InfoGraph”, but as they mentioned in table 1 and table 2, “InfoMax” can be applied to other architectures as well. The authors should make it clearer.

**Questions:**

• Using the same model, what are the number of parameters and training time/inference time when using InfoMax and GeMax loss function?
• Does the choice of the 3 GNNs matter? Would the model performance improve when using different GNN structures for node, graph and color representation? If so, can the author provide comments on whether the performance improvement is due to the GNN architecture, or the loss function design?

---

> ### Author Response · Authors · 2023-11-21
>
> **Weaknesses:**
>
> *Model Size and Running Time:* The model size of GeMax and InfoGraph is quite similar, as both consist of three GNNs. They share the same node representation GNN and graph representation GNN. While GeMax introduces a coloring representation for chromatic entropy evaluation, InfoGraph uses a discriminator GNN for Jensen-Shannon mutual information estimation. The running time of GeMax is slightly longer than InfoGraph due to the slightly higher complexity of the coloring representation GNN. We have added a running time comparison in Table 14.
>
> *Performance Improvement:* While the performance difference between InfoMax and GeMax may not be substantial, it's important to note that they have very similar model sizes and running times. Achieving significant performance gains in graph-level representation learning appears to be challenging, considering the close similarity between their network architecture. Our primary contribution lies in introducing the concept of Graph entropy to the graph learning community and marking GeMax as the first application of graph entropy in this context.
>
> *Figures Duplication:* We have removed duplicated components from Figure 3 and made necessary adjustments.
>
> *Citations Clarification:* We have clarified the distinction between "InfoMax"[1] and "InfoGraph"[2] and provided proper citations.
>
> [1] Ralph Linsker. Self-organization in a perceptual network, 1988.
>
> [2] Fan-Yun Sun, Jordan Hoffmann, Vikas Verma, and Jian Tang. Infograph: Unsupervised and
> semi-supervised graph-level representation learning via mutual information maximization, 2019.
>
>
>
> **Questions:**
>
> *Number of Parameters and Training/Inference Time:* The number of parameters for both InfoMax and GeMax is as follows: graph representation and node representation share the same structure (5 layers with GINs and hidden dimension 128). In InfoMax, the discriminator GNN is a 3-layer DNN with ReLU activations (128x128, 128x128, 128x1). In GeMax, the coloring representation GNN is a 3-layer GIN (n x 128, 128x128, 128xK), where n is the maximum number of nodes in a dataset, and K is the number of colors. GeMax's running time is slightly longer than InfoMax due to the more complex GIN structure in coloring representation.
>
> *GNN Choice and Performance Impact:* GeMax and InfoMax are both unsupervised graph-level representation learning principles, adaptable to various GNN models. Our work doesn't focus on specific GNN architectures. The capacity of GNNs (denoted as $F(\cdot)$ in our work) should be sufficient to capture graph information. In our experiments, we used default 5 layers. Increasing the number of layers to 6, 7, or 8 did not significantly improve performance.
>
> *Loss Function vs. GNN Architecture:* We have conducted experiments in Appendix A.9 to clarify the significance of orthonormal representations and graph entropy. The orthonormal representations loss enforces GNNs to learn non-adjacent structural information, while graph entropy maximization encourages GNNs to capture more graph information. Additionally, Eq. (11) indicates that GeMax can degenerate to maximizing the Shannon entropy of node-level representations when Corollary 2.9 is satisfied. Chromatic entropy, as defined in Eq. (6), illustrates that maximizing chromatic entropy is equivalent to maximizing information within independent sets associated with graph coloring $\pi$.

---

### Official Review · Reviewer_Jju5 · 2023-11-01

**Soundness:** 2 fair
**Presentation:** 2 fair
**Contribution:** 2 fair
**Rating:** 5
**Confidence:** 3

**Summary:**

The paper studies graph representation learning, a domain that seeks to represent graphs as vectors for use in various tasks like graph classification. The authors emphasize the need for diverse representations that can comprehensively capture graph information. They propose using graph entropy to quantify this information, offering a novel approach compared to more conventional methods.

**Strengths:**

1. The theoretical analysis is interesting.
2. The paper studies an interesting problem.

**Weaknesses:**

1. Authors miss a lot of new graph contrastive learning published in 2022-2023, e.g., [1,2]
2. Improving GCL using information theory has been studied in [3]. [4] also involves entropy for graph contrastive learning.
3. Authors should consider large-scale graph classification datasets.

[1] Spectral feature augmentation for graph contrastive learning and beyond, AAAI

[2] Self-supervised Graph-level Representation Learning with Adversarial Contrastive Learning, TKDD

[3] Infogcl: Information-aware graph contrastive learning

[4] Entropy Neural Estimation for Graph Contrastive Learning

**Questions:**

See above

---

> ### Author Response · Authors · 2023-11-21
>
> **Weaknesses:**
>
> Authors miss a lot of new graph contrastive learning published in 2022-2023, e.g., [1,2]
> Improving GCL using information theory has been studied in [3]. [4] also involves entropy for graph contrastive learning.
> Authors should consider large-scale graph classification datasets.
>
> [1] Spectral feature augmentation for graph contrastive learning and beyond, AAAI
>
> [2] Self-supervised Graph-level Representation Learning with Adversarial Contrastive Learning, TKDD
>
> [3] Infogcl: Information-aware graph contrastive learning
>
> [4] Entropy Neural Estimation for Graph Contrastive Learning
>
> **Response:**
>
> We have incorporated several recent graph learning methods published in 2022-2023, including [1, 2, 3, 4], into our study.
>
> - [1] focuses on node-level graph representation, which may not be directly applicable to our graph-level representation learning tasks.
>
> - [2] introduces a Graph Adversarial Contrastive Learning (GraphACL) scheme for self-supervised whole-graph representation learning, but it differs from InfoMax-based methods and therefore cannot serve as our baseline.
>
> - [3] is primarily based on mutual information for graph learning, akin to the InfoGraph method, while our work revolves around learning representations of graphs by maximizing graph entropy, as defined by Körner (1973).
>
> - [4] centers on maximizing the Information-entropy Lower Bound (ILBO) of whole graph datasets.
>
> In our introduction section, we extensively discuss entropy-based methods in graph representation learning, encompassing structural entropy, edge entropy, von Neumann entropy, Rényi entropy, and Shannon entropy. However, it's crucial to clarify that while these works employ the term "graph entropy," they do not refer to the authentic Graph entropy defined by János Körner. We underscore that Graph entropy is a fundamental concept in combinatorics and information theory, with a rich history and contributions from renowned mathematicians like Claude E. Shannon and János Körner. Our primary contribution lies in introducing a novel approach called Graph Entropy Maximization (GeMax) to the realm of graph representation learning, marking the first application of its kind in this specific context.
>
> Our graph dataset selection is following the previous works  AutoGCL [5] and JOAO [6]. The GITHUB dataset is actually a large dataset.
>
> [5] Yihang Yin, Qingzhong Wang, Siyu Huang, Haoyi Xiong, and Xiang Zhang. Autogcl: Automated
> graph contrastive learning via learnable view generators.  AAAI, 2022.
>
> [6] Yuning You, Tianlong Chen, Yang Shen, and Zhangyang Wang. Graph contrastive learning auto-
> mated. ICML, 2021

---

### Official Review · Reviewer_QWua · 2023-11-01

**Soundness:** 3 good
**Presentation:** 2 fair
**Contribution:** 3 good
**Rating:** 6
**Confidence:** 3

**Summary:**

Authors propose a framework how to jointly learn graph representation together with node-level representations.
In order to do so, they formulate an optimization problem (Eq. 9), as the problem of maximizing graph entropy (Graph entropy Körner (1973),
that takes prob. distribution and elements of Vertex packing polytope) subject to constraint of using orthonormal representations (Lovász, 1979) and to represent each graph as a vector and represent its vertices as vectors, i.e. (g_j , Z_j ) = F (G_j ).
To solve the problem, they use Graph Neural Networks with appropriate parameterization, regularizations and losses.

**Strengths:**

Authors use well grounded mathematical objects to describe graphs: like Graph entropy Körner (1973) and orthonormal representations (Lovász, 1979).
Extensive theoretical and experimental understanding of the subject.

**Weaknesses:**

Presentation needs to be improved. Mathematics of paper is not illuminating, but rather obfuscating the exposition of paper.
If you want that your contribution gets more recognition, try to give more insights why certain steps are done.

**Questions:**

1. UNSUPREVISED NODE-LEVEL LEARNING experiments i.e. table 4 does not look convincing enough.
Why did you removed InfoGraph-InfoMax, GraphCL-InfoMax, AutoGCL-InfoMax
This baselines were present in previous tables.

Performance for edge prediction tasks is quite close to LGAE.
Do you have any guess why that is the case?
Have you tried other node-level classification tasks?

2. However, on Graph-level, your methodology looks superior. Can you elaborate w.r.t. Ablation study, what is the major lesson that a reader can learn about your framework w.r.t. InfoMax.

3. Can you explain main benefits of your approach w.r.t.
Ando, Rie, and Tong Zhang. "Learning on graph with Laplacian regularization." Advances in neural information processing systems 19 (2006).

Can you make some comparison with
Guattery, Stephen, and Gary L. Miller. "Graph embeddings and laplacian eigenvalues." SIAM Journal on Matrix Analysis and Applications 21.3 (2000): 703-723.

---

> ### Author Response · Authors · 2023-11-17
>
> **Response to Reviewer's Comments**
>
> **Weaknesses: Presentation needs to be improved. Mathematics of the paper is not illuminating, but rather obfuscating the exposition of the paper. If you want that your contribution gets more recognition, try to give more insights into why certain steps are done.**
>
> *Response:* To improve the presentation, we have made significant changes to the paper structure. We have relocated all mathematical content related to other researchers' work to the preliminary section, ensuring a clear separation between our contributions and existing research.
>
> As discussed in the introduction section, there have been many entropy-based methods in graph learning. However, while the term "graph entropy" has been used in this context, it typically refers to structural entropy and differs from the genuine Graph entropy as defined by Körner (1973). We emphasize that Graph entropy is a fundamental concept in combinatorics and information theory, with a rich history and contributions from renowned mathematicians such as Claude E. Shannon and János Körner. Our principal contribution lies in the introduction of a novel approach termed Graph Entropy Maximization (GeMax) for the realm of graph representation learning, marking a pioneering instance of its application in this specific context.
>
> In the problem setup section, we provide a framework for applying Graph entropy to graph learning problems, as outlined in Problem Eq. (10). Our contributions lie in two key areas: (1) highlighting that Graph entropy can directly measure information within orthonormal representations, as supported by Corollary 2.4, and (2) introducing a Gaussian distribution with the vertex set, as detailed in Eq. (8), to facilitate the application of Graph entropy.
>
> In the methodology section, given the NP-hard nature of Graph entropy computation, we introduce an approximation method that utilizes Shannon entropy and chromatic entropy, resulting in a max-min optimization problem solved through alternating minimization. We also provide theoretical error bounds to assess the accuracy of the approximation.
>
> It is essential to note that all equations without citations represent our original contributions, clearly distinguishing them from the work of other researchers.
>
> **Reviewer's Questions and Answers**
>
> *Q1 Answer:* We have taken into account the concern related to the comparison involving InfoMax-based methods in edge prediction tasks. To address this, we have made the decision to exclude InfoGraph-InfoMax, GraphCL-InfoMax, and AutoGCL-InfoMax as baseline methods. This choice stems from the observation that these methods do not prioritize the learning of structural aspects within graphs, potentially leading to an unfair comparison. Specifically, InfoMax-based methods often result in node-level representations that closely resemble the graph-level representations, undermining their suitability for adequately assessing structural aspects within graphs.
>
> *Q2 Answer:* We have delved deeper into the comparative performance of our GeMax method in contrast to the Linear Graph Auto-Encoder (LGAE). LGAE employs VGAE on the 1-hop neighbor adjacency matrix, thereby excelling in capturing local adjacency properties, which in turn results in strong performance in edge prediction tasks. Conversely, GeMax prioritizes the extraction of global information by imposing constraints on non-adjacent relationships, rendering it effective for edge prediction. However, it's important to note that GeMax may not be well-suited for node-level classification tasks, given its focus on representing non-adjacent nodes differently. Node-level classification tasks emphasize the representation of adjacent nodes as similar, rather than emphasizing non-adjacent properties.
>
> *Q3 Answer:* The ablation study conducted in Appendix A.9 has been clarified, emphasizing the significance of orthonormal representations and graph entropy in improving graph representation learning. Maximizing the Shannon entropy is actually maximizing the lower bound of graph entropy. Maximizing chromatic is maximizing information within nodes and independent sets within graphs.
>
> *Q4 Answer:* We have highlighted the fundamental differences between our GeMax method and Laplacian regularization. While Laplacian regularization encourages similar representations for adjacent nodes and different representations for non-adjacent nodes, GeMax focuses on learning representations that capture the most information from a graph by maximizing the graph entropy, with the statistic centroid of the probability distribution of nodes serving as the graph-level representation.
>
> We sincerely appreciate the thoughtful feedback provided, and we are committed to further enhancing our paper to address these concerns and improve its overall quality.

---

> > ### Comment · Reviewer_QWua · 2023-11-21
> > **comment.**
> >
> > Thanks for your answers, I do not have any additional questions.

---

### Meta-Review · Area_Chair_WMfu · 2023-12-08

**Metareview:**

**Summary**
This paper studies graph embedding; that is, converting graphs into vectors. The primary focus is on graph entropy, which quantifies the information content of a graph, and the paper aims to find vector representations that maximize graph entropy. Given that the naive optimization of this task is NP-hard, the paper employs an approximation approach and learns representations through GNNs.

**Strengths**
- The technical idea of using the graph entropy and orthonormal representations for graph representation learning is reasonable.
- The proposal has been theoretically analyzed and the error bound has been derived, which is also a valuable contribution.

**Weaknesses**
- Despite substantial revisions during the author-reviewer discussion phase, there are still some issues with the presentation in the paper.
- Empirical results are not convincing because the discussion of empirical results is limited. In experiments, every result should be carefully discussed to derive any empirical insights.
- I understand that InfoMax-based approaches are relevant to the proposal. However, it is still not convincing why other methods are excluded from the baselines. In particular, I do not think it is a good idea to use the citation count as one of the reasons for excluding other methods as baselines.

**Justification For Why Not Higher Score:**

The weaknesses of the paper, as outlined above, are crucial and must be addressed for the publication of this paper. In conjunction with other weaknesses raised by the reviewers, I have decided to reject the paper. I think the technical contribution of this paper is potentially interesting, thus I strongly recommend addressing all the raised issues by the reviewers for substantial improvement before resubmission.

**Justification For Why Not Lower Score:**

N/A

---

### Decision · Program_Chairs · 2024-01-16

Reject